# An interphase actin wave promotes mitochondrial content mixing and organelle homeostasis

Stephen M. Coscia[1,2,3], Andrew S. Moore[4], Cameron P. Thompson[1,2,5], Christian F. Tirrito ®[6,7], E. Michael Ostap[1,2] & Erika L. F. Holzbaur ®[1,2] ✉

Across the cell cycle, mitochondrial dynamics are regulated by a cycling wave of actin polymerization/depolymerization. In metaphase, this wave induces actin comet tails on mitochondria that propel these organelles to drive spatial mixing, resulting in their equitable inheritance by daughter cells. In contrast, during interphase the cycling actin wave promotes localized mitochondrial fission. Here, we identify the F-actin nucleator/elongator FMNL1 as a positive regulator of the wave. FMNL1-depleted cells exhibit decreased mitochondrial polarization, decreased mitochondrial oxygen consumption, and increased production of reactive oxygen species. Accompanying these changes is a loss of hetero-fusion of wave-fragmented mitochondria. Thus, we propose that the interphase actin wave maintains mitochondrial homeostasis by promoting mitochondrial content mixing. Finally, we investigate the mechanistic basis for the observation that the wave drives mitochondrial motility in metaphase but mitochondrial fission in interphase. Our data indicate that when the force of actin polymerization is resisted by mitochondrial tethering to microtubules, as in interphase, fission results.

Cellular health is dependent on mitochondrial homeostasis, as mitochondria produce much of cellular energy in the form of ATP and also serve as hubs for cellular signaling[1]. The health of the mitochondrial network is maintained by the dynamic processes of fission and fusion[2]. Both processes are governed by GTPases, where fusion requires mitofusins and OPA1 while fission requires DRP1[3]. Accumulating evidence indicates that the actin cytoskeleton also plays a role in mitochondrial fission[4–6]. In a mechanism that provides spatial and temporal regulation of mitochondrial fission during interphase, a wave of transient filamentous (F-) actin assembly and disassembly cycles through mitochondrial subpopulations, promoting their fragmentation with subsequent re-fusion once the waves moves on (Fig. 1A,

Supplementary Movie 1)[7]. How this actin wave induces mitochondrial fission and for what purpose remains unclear.

In metaphase the mitochondrial actin wave no longer promotes fission but instead drives bursts of motility. While most metaphase mitochondria within the wave are completely enveloped in clouds of F-actin, stochastic symmetry breaking of these clouds yields mitochondrially-associated comet tails that drive randomly directed and transient mitochondrial translocation[8]. In canonical comet tail motility actin monomers incorporate into a larger F-actin network at the interface between this network and a cellular structure, producing force on this structure and displacing it, and this paradigm may apply here[9,10]. Upon inhibition of the metaphase actin wave, mitochondria

[1]Department of Physiology, University of Pennsylvania Perelman School of Medicine, Philadelphia, PA, USA. [2]Pennsylvania Muscle Institute, University of Pennsylvania Perelman School of Medicine, Philadelphia, PA, USA. [3]Cell and Molecular Biology Graduate Group, University of Pennsylvania Perelman School of Medicine, Philadelphia, PA, USA. [4]Howard Hughes Medical Institute, Janelia Research Campus, Ashburn, VA, USA. [5]Biochemistry and Molecular Biophysics Graduate Group, University of Pennsylvania Perelman School of Medicine, Philadelphia, PA, USA. [6]Raymond G. Perelman Center for Cellular and Molecular Therapeutics, The Children's Hospital of Philadelphia, Philadelphia, PA, USA. [7]Department of Pathology and Laboratory Medicine, University of Pennsylvania, Philadelphia, PA, USA. ✉e-mail: holzbaur@pennmedicine.upenn.edu

are no longer spatially shuffled as they are in control cells. Mitochondria within single cells can be heterogeneous[11], differing in DNA sequence, protein composition, and level of oxidative damage[12,13]. As such, local sub-populations of mitochondria can arise via clonal expansion, and for truly symmetric cell division these sub-populations must be dispersed prior to cytokinesis[14]. Thus, the spatial shuffling induced by the cycling actin wave promotes the symmetrical inheritance of diverse mitochondria by daughter cells.

While the metaphase actin wave serves to spatially mix mitochondria, we wondered whether the interphase wave could mix mitochondria in a different way. Namely, the interphase wave could promote mitochondrial content mixing, a process in which heterogeneous mitochondria share components via fusion. While wave-associated F-actin assembly induces mitochondrial fission, the subsequent disassembly of mitochondrially-associated F-actin enhances re-fusion. During this period, fragmented organelles might re-fuse with neighbors originating from distinct mitochondria. Supporting this possibility, we have observed instances of such 'hetero-fusion'[7], but this possibility has not yet been investigated in detail. Mitochondria within a given cell can differ significantly in regard to respiratory capacity, and fusion with healthy mitochondria can rescue the functioning of damaged organelles in a process known as complementation[15]. Thus we hypothesize that actin wave-driven complementation might contribute to the overall health of the mitochondrial network.

Mitochondria exhibit cell cycle-induced changes in interactions with microtubules: in interphase, mitochondria are closely associated with the microtubule cytoskeleton via the molecular motors kinesin-1 and cytoplasmic dynein, but these associations are dissolved during metaphase by phosphorylation of motor complex components[16,17]. We wondered whether the different outcomes of the wave: mitochondrial fragmentation during interphase and mitochondrial motility during metaphase, might result from these changes in cytoskeletal association. We test the hypothesis that the fragmentation of mitochondria in interphase is dependent on a close association of mitochondria with microtubules.

Here we explore the mechanisms underlying actin wave-mediated mitochondrial fragmentation. We identify formin-like protein 1 (FMNL1) as an essential F-actin nucleator/elongator required for the wave, likely functioning downstream of cyclin-dependent kinase 1 (CDK1), and then use depletion of FMNL1 as a specific tool to uncover a role for the interphase actin wave as a positive regulator of mitochondrial health. Next, we probe the basis for this finding, and demonstrate that the interphase wave fragments mitochondria in a spatially- and temporally-restricted manner that facilitates the subsequent hetero-fusion of mitochondrial fragments. Finally, we demonstrate that microtubule depolymerization is sufficient to block actin wave-induced mitochondrial fission, suggesting that the interphase wave promotes fragmentation by producing force on mitochondria that is resisted by their tethering to microtubules, resulting in tubulation that can promote DRP1-dependent fission. Strikingly, we find that in the absence of the resistive force provided by microtubule binding, the interphase actin wave can induce comet tail propulsion of mitochondria, thus behaving similarly to the metaphase wave.

## Results

### FMNL1 is required for the mitochondrial actin wave

HeLa cells expressing Lifeact-GFP and mito-DsRed2 to label actin and mitochondria, respectively, were examined by live imaging to visualize the propagation of the interphase actin wave and associated mitochondrial fission (Fig. 1A & Supplementary Movie 1)[7]. The propagation of this cycling wave is effectively inhibited by depletion of subunits of the F-actin-nucleating ARP2/3 complex or by pharmacological inhibition of the complex using CK-666[7,8]. However, given the ARP2/3

complex functions in many different cellular contexts[18], we sought a tool that could more specifically ablate the actin wave in order to more accurately assess the role of the wave in the maintenance of mitochondrial homeostasis during interphase. Our previous work suggested that the actin wave is dependent on a member of the formin family of actin polymerizing proteins as the pan-formin inhibitor SMIFH2 dramatically reduced the area of the wave[7,19], although we note concerns about the specificity of this reagent[20]. There are 15 human formins[21] but we focused on members of the FMNL subfamily, which have been implicated in the assembly of cortical F-actin and, importantly, are known to be phosphorylated by CDK1[22-25]. CDK1 inhibition may reduce the size of the actin wave[26].

We used siRNAs to individually deplete FMNL1, 2, or 3 from HeLa cells and we stained knockdown and control cells for the mitochondrial marker TOMM20 and used phalloidin to label F-actin for examination by confocal microscopy (Fig. 1B). In a blinded analysis we measured actin wave size by drawing a perimeter around clustered, actin-positive mitochondria (Fig S1A). Such methodology faithfully reports on the size of the wave, as other described actin-mitochondria interactions are generally not apparent in single-timepoint fluorescence micrographs of unperturbed cells[27]. Depletion of FMNL1 completely inhibited the interphase actin wave whereas siRNAs directed against either FMNL2 or 3 had no effect (Fig. 1C); a similar trend was observed when considering the percentage of cells that feature a wave (Fig. 1D). Of note, we assessed knock-down efficiency by western blotting and qPCR (Fig. 1E & Fig S1B), finding that we were able to deplete, on average, 70% of endogenous FMNL1, 83% of endogenous FMNL2, and 84% of endogenous FMNL3 transcripts. Next we examined the effects of FMNL1 depletion in metaphase cells and again noted a striking ablation of the actin wave (Fig S1C, D). However, we also noted that the original FMNL1 siRNA induced chromosomal misalignment in some metaphase cells, which were excluded from analysis. To confirm our findings, we used an independent siRNA targeted against FMNL1 (siRNA #2), which effectively ablated the wave in both interphase and metaphase cells without causing chromosomal misalignment (Fig S1E–G) and induced 73% depletion of endogenous FMNL1 (Fig S1H).

We next assessed the effect of FMNL1 knockdown on actin wave size in live cells. To this end we transfected HeLa cells with our initial FMNL1 siRNA, which we utilized throughout the remainder of this study as it yielded consistent knockdown without demonstrating off-target effects in interphase cells. Cells were co-transfected with a plasmid encoding Lifeact-GFP and then stained with the mitochondrial dye MitoTracker Deep Red FM (Fig S1I). In time-lapse recordings of interphase cells we observed robust wave propagation in control cells while FMNL1-depleted cells featured diminished actin waves, with maximum wave area reduced from $158\,\mu m^2$ to $39\,\mu m^2$, comparing medians (Fig S1J). Further, in random timepoint sampling of many cells, we confirmed that the actin wave was ablated with FMNL1 siRNA treatment (Fig S1K).

In order to test the generalizability of our findings we treated a second cell line, COS-7 cells, with FMNL1 siRNA prior to expressing Lifeact-GFP and mito-DsRed2 and then used confocal microscopy random timepoint images to measure actin wave size. We found this metric to be greatly decreased in FMNL1 knock-down cells compared to control COS-7 cells (Fig S1L, M), just as in HeLa cells. We used qPCR to confirm efficient depletion of FMNL1, to 35% of endogenous levels (Fig S1N).

Finding a requirement for FMNL1 for the actin wave, we then sought to understand if FMNL1 is recruited to wave-positive mitochondria. We expressed in HeLa cells GFP-FMNL1 and stained for mitochondria using a TOMM20 antibody and F-actin using phalloidin, and then visualized the cells using confocal microscopy. We noted FMNL1 recruitment to actin wave-positive mitochondria, which we confirmed with line scan analysis (Fig. 1F).

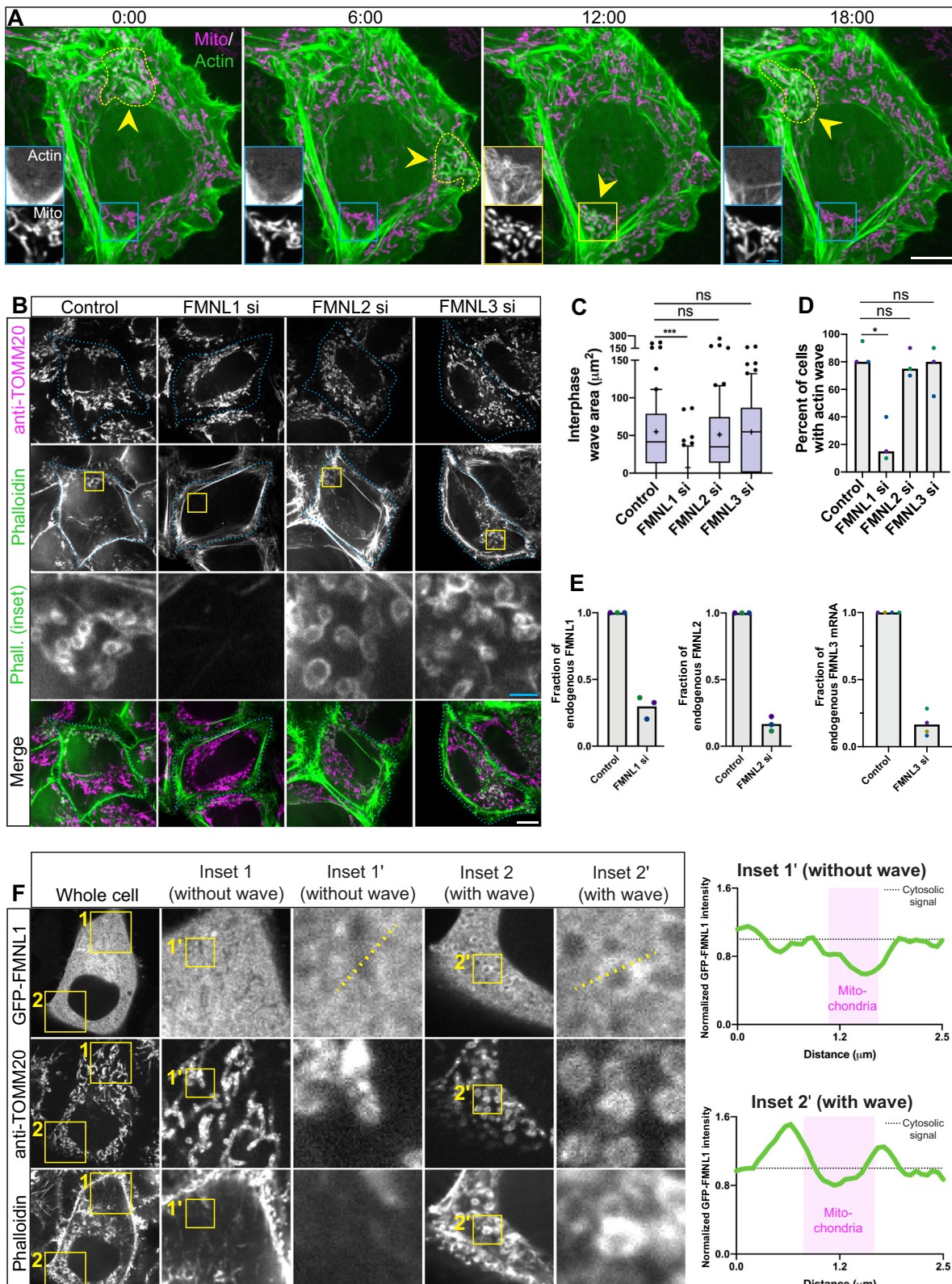

## A CDK1-FMNL1 axis positively regulates the mitochondrial actin wave

CDK1 has been implicated in positively regulating the actin wave, as actin cycling is blocked by the CDK1 inhibitor Ro-3306 in cells arrested in mitosis[26]. We wondered if a downstream effector of CDK1 is FMNL1, as there is a consensus CDK1 site in FMNL1 (Fig. 2A), and the analogous site in the related protein FMNL2 has been shown to be phosphorylated by CDK1[25]. Further, a proteomics study concluded that this site in FMNL1 is phosphorylated in a CDK1-dependent manner[28], and multiple entries in the curated database of phosphorylated peptides, PhosphoSitePlus, indicate the same[29]. Indeed, in a more targeted mass spectrometry experiment on immunoprecipitated GFP-FMNL1 we were able to detect this phosphorylation (Fig. 2B). To more directly test if CDK1 is the responsible kinase, we treated HeLa cells expressing

**Fig. 1 | FMNL1 is required for the actin wave. A** Timelapse images of interphase actin wave in HeLa cells; F-actin and mitochondria marked with Lifeact-GFP and mito-DsRed2, respectively. Yellow arrow and dotted line indicate the position of the wave. Insets are displayed with enhanced contrast for ease of viewing. Scale bar 10 μm for whole cell images, 2 μm for insets. **B** Representative images of phalloidin (F-actin marker) and anti-TOMM20 (mitochondrial marker) in interphase cells treated with siRNAs to either FMNL1, 2 or 3. Scale bar 10 μm for whole cell images and 2 μm for insets. Cell boundaries are outlined in cyan. **C** Quantitation of actin wave size for each group in interphase cells, where Whiskers represent 10–90th percentile, center lines indicate medians, and plus signs indicate means. $n = 60$ across 3 biological replicates. **D** Percent of cells displaying an actin wave, bars represent medians. $n = 3$ biologically independent samples. **E** Quantitation from

western blot and qPCR experiments examining the knock-down efficiencies of FMNL1, 2 and 3, siRNAs. Bars represent means. $n = 3$ and 4 biologically independent experiments for western blot and qPCR experiments, respectively.
**F** Representative images of GFP-FMNL1 recruitment to actin wave-positive mitochondria, where F-actin is marked by phalloidin and mitochondria are marked by a TOMM20 antibody. Whole cell image scale bar is 10 μm, medium zoom inset scale bar is 5 μm, and high zoom scale bar is 1 μm. Experiment repeated 3 times with similar results. Accompanying are line scans where all signal is normalized to cytosolic signal. **A–F** Differently colored points indicate different biological replicates. Statistical test used was Kruskal-Wallis test with Dunn's multiple comparisons. Source data are provided as a Source Data file. *, **, and *** indicate p-values of < 0.05, ≤ 0.005, and ≤ 0.0005 respectively.

GFP-FMNL1 with either vehicle or the drug adavosertib, which is an inhibitor of the CDK1 negative regulator WEE1 and thus should effectively enhance CDK1 activity[30]. We pulled-down GFP-FMNL1 from these cells, and used western blots to probe these samples with an antibody specific for the phosphorylated CDK1 consensus sequence. Pull-down samples from cells treated with adavosertib showed a consistently higher signal when probed with the phosphorylated CDK1 consensus sequence antibody compared to control cells (Fig. 2C), further supporting the hypothesis that FMNL1 is a substrate of CDK1.

To gain a clearer understanding of the role of CDK1 activity in regulating the actin wave, we treated HeLa cells expressing Lifeact-GFP with either vehicle or adavosertib, labeled mitochondria with the dye MitoTracker Deep Red FM, and visualized interphase cells using random-timepoint confocal microscopy. We did not record a difference in either wave area or percent of cells featuring a wave (Fig S2A). We then recorded videos of cells expressing Lifeact and measured the speed of the actin wave in a blinded analysis (Fig. 2D, G, Supplementary Movie 2). We found that treatment with adavosertib increased the speed of the wave significantly, from 3.7 μm/min to 5.6 μm/min on average. Next we treated cells stably expressing Lifeact-GFP and mito-DsRed2 with two different CDK1 inhibitors, Ro-3306 and CGP74514A[31,32], and used confocal microscopy to image interphase cells (Fig. 2E, H, Fig S2B). The CDK1 inhibitors were tested as part of a larger screen; the DMSO control data run in parallel with this screen were previously published[8]. We found that either drug strongly inhibited the wave considering wave size and percent of cells featuring a wave. Finally, to further probe the effects of CDK1 downregulation, we treated HeLa cells with an siRNA targeting CDK1, stained for the mitochondrial marker TOMM20 and F-actin using phalloidin, and used confocal microscopy to examine interphase cells. We found that the CDK1 siRNA led to a strong reduction in the size of the actin wave (Fig. 2F, I) and western blotting indicated this siRNA decreased endogenous CDK1 levels by 46% (Fig. 2J & Fig S2C). These results are consistent with CDK1 activity positively regulating the actin wave.

To test if CDK1 regulates the actin wave via phosphorylation of FMNL1, we mutated the serine residue within the consensus CDK1 site in FMNL1 to an alanine residue to form a non-phosphorylatable mimetic. Then we overexpressed either GFP alone, WT GFP-FMNL1, or GFP-FMNL1 S1031A in HeLa cells, stained with TOMM20 antibody and phalloidin, and visualized the samples using confocal microscopy (Fig. 2K). Interphase cells expressing FMNL1 S1031A exhibited smaller actin waves than were observed in cells expressing either GFP or WT FMNL1, consistent with a dominant negative mechanism (Fig. 2L & Fig S2D). Neither of the FMNL1 constructs utilized appeared to undergo meaningful degradation, as assessed by western blot (Fig. 2M), and these FMNL1 constructs were expressed to similar extents in the cells analyzed (Fig. 2N). We then performed the converse experiment, where we mutated the serine residue within the consensus CDK1 site to a glutamate residue to generate a phosphomimetic, and assessed the effects of overexpression (Fig. 2O). We randomly picked cells to image and conducted three independent

biological replicates to obtain a sample size of 60 per group, analyzing imaged cells with no exclusions. Overexpression of the phosphomimetic construct increased actin wave size (Fig. 2P) but did not influence either the percent of cells with a wave or wave speed in cells that were expressing Lifeact-mScarlet and stained with Mito-Tracker Deep Red FM (Fig S2E). Further, western blotting for GFP-FMNL1 S1031E did not indicate meaningful degradation of the construct (Fig. 2Q), and an analysis of cells from our fixed imaging indicated the different groups expressed WT and S1031E FMNL to the same extent (Fig. 2R). Together, these results support the model that CDK1 positively regulates the actin wave by phosphorylating FMNL1.

## The interphase actin wave maintains mitochondrial homeostasis

With the ability to ablate the actin wave in a more specific manner by FMNL1 depletion rather than inhibition of the ARP2/3 complex, we next tested the hypothesis that the interphase wave maintains mitochondrial homeostasis. To do so we treated HeLa cells with FMNL1 siRNA and also expressed the mitochondrial marker mito-sBFP2 before loading cells with the mitochondrial membrane potential sensitive dye TMRE for live imaging (Fig. 3A)[33]. To analyze the resulting data we segmented the mitochondrial networks of interphase cells and measured the average TMRE intensity within this area. TMRE intensity was decreased by 43% on average in HeLa cells lacking FMNL1 (Fig. 3B). We performed a parallel experiment in COS-7 cells and noted a 62% decrease in TMRE intensity (Fig S3A), demonstrating that the effects of FMNL1 depletion are not limited to HeLa cells. Similarly, a significant decrease in mitochondrial membrane potential was observed in HeLa cells treated with ARP3 siRNA, an approach that also effectively inhibits the actin wave albeit less specifically than FMNL1 depletion. As a control, we examined mitochondrial TMRE intensity in cells following depletion of the protein WHAMM, which assists the ARP2/3 complex in generating F-actin in certain contexts but is not required for the actin wave[8], and found no change. We confirmed our knock-downs by western blot, and found that the ARP3 and WHAMM siRNAs were efficient at depleting their targets, achieving 83% and 84% knock-down, respectively (Fig. 3F & Fig. S3B).

Next, we investigated the influence of FMNL1 on mitochondrial oxygen consumption. We treated HeLa cells with FMNL1 siRNA and applied a mitochondrial stress test on a Seahorse Bioanalyzer. The results indicated a deficit in basal mitochondrial oxygen consumption rate (OCR) compared to controls – specifically, the average OCR/μg protein ratio for control cells was 3.1 pmol/min whereas for FMNL1 knock-down cells we measured 1.6 pmol/min (Fig. 3C).

Finally, to assess reactive oxygen species (ROS) levels in cells with and without an active wave, we added the ROS sensor CellROX, as well as the cell membrane dye CellMask, to FMNL1-depleted interphase cells and control cells and then performed live imaging (Fig. 3D). CellROX intensity was increased by 43% on average after FMNL1 depletion and wave ablation. ARP3 depletion had an even stronger effect – specifically, a 131% increase in CellROX over control cells

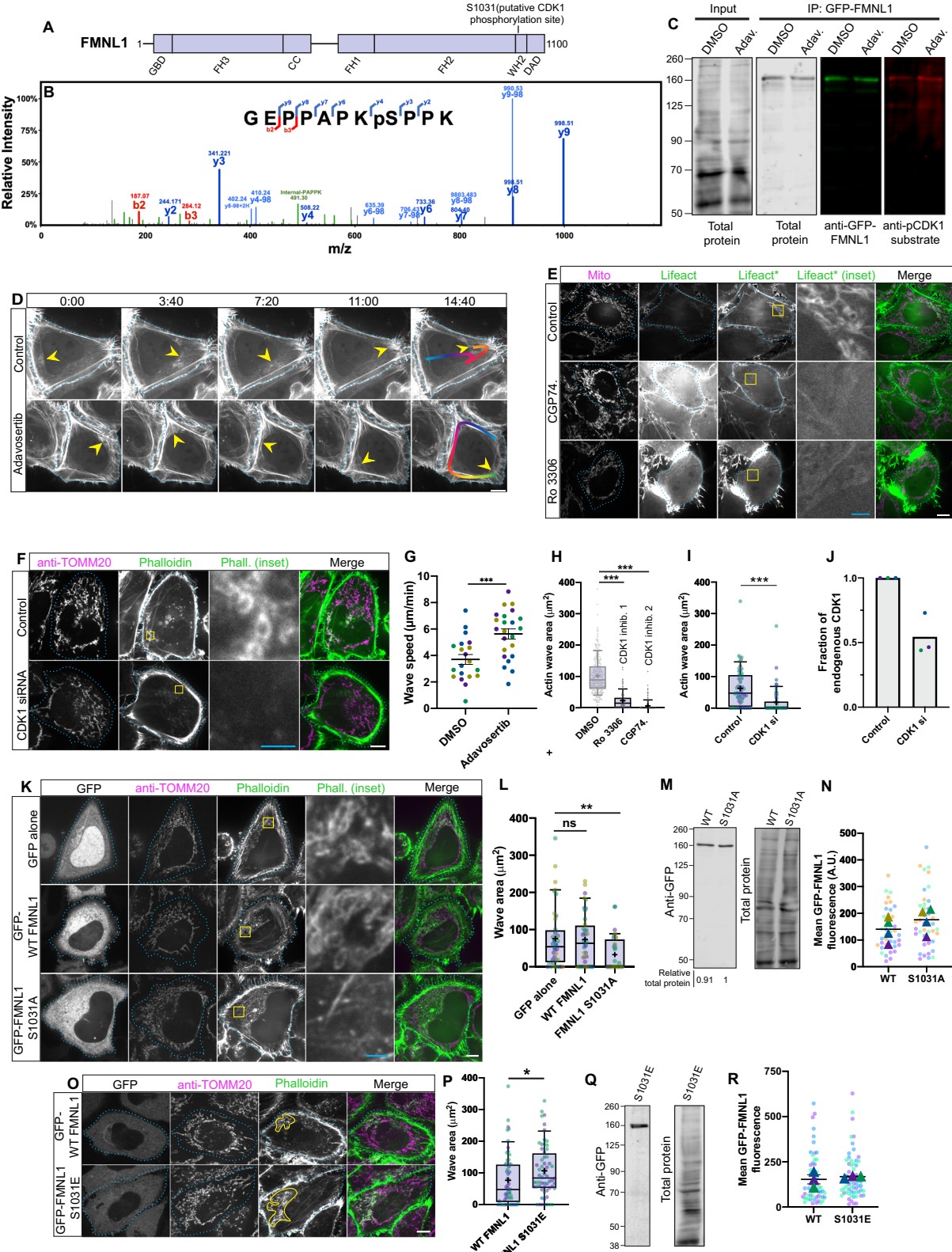

(Fig. 3E). Again, in our control experiment, we saw no change in Cell-ROX intensity for WHAMM-depleted cells.

FMNL1 was recently shown to be required for transient F-actin accumulation on newly and fully depolarized mitochondria[34], the only established link between FMNL1 and mitochondria to date, so we asked if this phenomenon might contribute to the phenotype of decreased mitochondrial health that we observed with FMNL1

knockdown. We expressed mito-SNAP and Lifeact-mEmerald in HeLa cells, then loaded the cells with the membrane potential-sensitive dye TMRE and imaged interphase cells live, recording the frequency of spontaneous mitochondrial depolarization events and the fraction of these events that were actin positive (Fig S3C). Under these conditions, spontaneous mitochondrial depolarization events were rare and only a small fraction of these were associated with local accumulation of actin

**Fig. 2 | CDK1 positively regulates the actin wave via FMNL1. A** FMNL1 schematic
(**B**) MSMS scan of residues 1024–1034 (1 experiment). **C** Representative GFP-FMNL1
IP from cells treated with vehicle (~2 h DMSO) or adavosertib (~6.5 h 300 nM).
Probed with anti-GFP and anti-phoso-CDK1 substrate sequence. Repeated 3X.
**D** Effects of ~4 h adavosertib at 300 nM on wave. Unequal contrasting between
groups for ease of viewing. **E** Effects of CDK1 inhibitors on wave. Asterisk denotes
contrast adjustment between conditions, performed as acquisition parameters
varied throughout experiment. **F** CDK1 siRNA treatment. **G** Effects of adavosertib
on wave speed; bars represent means with SEM. Two-sided Student's t-test. $N = 20$
and 23 cells across 4 experiments for DMSO and Wee1 inhibitor, respectively.
**H** Effects of CDK1 inhibitors on wave area. CDK1 inhibitors tested in a larger screen;
DMSO data run in parallel previously published in Extended figure 8 A of ref. [8].
Analyzed by Kruskal-Wallis test with Dunn's multiple comparisons was run. $n = 151$
and 212 cells across 3 experiments for Ro3306 and CGP74 conditions, respectively.
**I** Effects of CDK1 knockdown on wave area. Kruskal-Wallis test with Dunn's multiple

comparisons. $n = 60$ cells across 3 experiments. **J** CDK1 knockdown (see Figure S2C). Means from 3 experiments. **K** Effects of GFP-FMNL1 WT and S1031A on
wave; staining TOMM20 and phalloidin. GFP signal for 'GFP alone' condition was
differentially scaled. **L** Wave areas analyzed by Kruskal-Wallis test, Dunn's multiple
comparisons; 40 cells across 4 experiments. **M** Western blot of GFP-FMNL1 WT and
S1031A. **N** Mean fluorescence: GFP-FMNL1 WT/S1031A. Bars indicate means of triangles, which represent means of 4 biological replicates. **O** Effects of S1031E on
wave; staining for TOMM20 and phalloidin. **P** Effects on wave area; two-sided
Mann-Whitney test, 60 cells across 3 experiments. **Q** Western of GFP-FMNL1 S1031E.
**R** Mean fluorescence: GFP-FMNL1 WT/S1031E. **A–R** Scale bars: 10 μm for cell images, 2 μm for insets. For blots, antibody/total protein approximately aligned/scaled
the same; kDa, unit of molecular weight. Box plots: center lines indicate medians,
plus signs indicate means, error bars = 10–90th percentile. Source data provided.
*, **, and *** indicate p-values of < 0.05, ≤ 0.005, and ≤ 0.0005 respectively.

(4%). Thus we conclude that under the conditions of our assays, the
primary consequence of FMNL1 depletion is the ablation of the actin
wave, which is sufficient to impair mitochondrial health.

## Mitochondrial health may not feedback to control the actin wave

As our data indicate that the interphase actin wave contributes to the
maintenance of mitochondrial homeostasis, we next asked if there is a
mitochondria-intrinsic element to the regulation of the wave. Mitochondria are known to buffer cytosolic calcium, and calcium waves
have been observed in cells[35], so we tested if changes in peri-
mitochondrial calcium levels might correlate with the actin wave. To
do so we co-expressed the calcium sensor GCaMP6s and the actin
marker Lifeact-miRFP in HeLa cells, then stained cells with MitoTracker
Red CMXRos and recorded videos of interphase cells (Fig. 4A, Supplementary Movie 3). While robust propagation of the mitochondrial
actin wave was evident, we did not observe any changes in GCaMP6s
signal associated with the wave.

We next asked if mitochondrial polarization state feeds back to
influence the interphase actin wave. To test this we treated cells with
the drug CCCP, which permeabilizes the mitochondrial inner membrane to protons, rendering these organelles non-functional[36,37]. CCCP
addition to cells has been shown to induce transient accumulation of
F-actin on all mitochondria, peaking at ~2 min with disassembly
occurring ~7.5 min after drug addition in U2OS cells[38]. In HeLa cells, we
have observed actin accumulation on all mitochondria up to 15 min
after CCCP addition[7]. Here, we treated HeLa cells expressing Lifeact-
GFP and mito-sBFP2 with CCCP, waited 30 min, and acquired single
timepoint images of live interphase cells to assess the percent of cells
with actin enriched on all mitochondria, actin enriched on just a subset
of mitochondria, or no mitochondrial actin (Fig. 4B). We found that for
many cells either most or all mitochondria were actin free. Next we
conducted the same experiment but prior to CCCP addition we loaded
cells with the mitochondrial membrane potential-sensitive dye TMRE.
We recorded videos of cells that had lost their TMRE staining consistent with mitochondrial depolarization, and that also featured a
subpopulation of mitochondria enriched for actin (Fig. 4C, Supplementary Movie 4). In these cells actin waves propagated at the same
speed as in control cells (Fig. 4D). These results indicate that mitochondrial polarization state does not feed-back to influence the actin
wave, as the wave persists despite a strong mitochondrial insult like
CCCP treatment.

## The interphase actin wave promotes mitochondrial content mixing

Our data suggest the interphase actin wave maintains mitochondrial
function, so we next probed the underlying mechanism. We hypothesized that the cycling actin wave drives the short-lived local
fragmentation of mitochondrial subpopulations, which in turn gives

the resulting mitochondrial fragments the opportunity to re-fuse
with distinct neighbors, a process we refer to as mitochondrial 'hetero-fusion.' Hetero-fusion could facilitate complementation, in
which a damaged mitochondrion can be rescued by fusion with a
healthy mitochondrion that can share functional copies of mitochondrial DNA[15]. We co-transfected HeLa cells with Lifeact-GFP and
mito-DsRed2, recorded videos of those cells in interphase, and were
able to observe events of actin wave-associated mitochondrial
hetero-fusion (Fig. 5A)[7]. We also observed this phenomenon in COS-7
cells, indicating it might be generalizable to other cell types (Fig S4A).
To quantitatively test the hypothesis that the wave induces heterofusion, we expressed photo-activatable GFP (PA-GFP) fused to a
mitochondrial marker to enable the labeling of mitochondrial subpopulations in HeLa cells with 405 nm light and treated cells with
FMNL1 siRNA to ablate the actin wave prior to live imaging during
interphase (Fig. 5B). We then segmented the labeled mitochondria at
the beginning and end of the resulting movies and measured the
percent change in area. While a bounding area analysis would inform
on mitochondria dispersion, the percent change in area occupied by
PA-GFP-labeled mitochondria, which is agnostic to mitochondrial
motility, reflects mitochondrial fusion[39] as this metric increases
because labeled mitochondria share their contents with unlabeled
organelles. We found that FMNL1 knock-down substantially reduced
the percent increase in the area of labeled mitochondria, from 37% to
11%, on average. These findings suggest there is reduced mitochondrial fusion in the absence of the cycling actin wave, perhaps
because the wave is not driving the local and time-limited fragmentation of mitochondria in a manner that enhances stochastic
recombination with distinct neighbors. This result is unlikely to be
due to any non-actin wave alterations in the fission/fusion balance
brought about by FMNL1 knock-down, as we did not find this intervention to change gross interphase mitochondrial connectivity using
the analysis outlined in our previous work[40] on confocal images of
cells expressing Lifeact-GFP and stained with MitoTracker Deep Red
FM (Fig S4B). These data then support our hypothesis that the
interphase actin wave promotes mitochondrial content mixing.

To test this model more directly, in a separate experiment we
expressed Lifeact-mScarlet and mito-mEmerald in HeLa cells, and used
confocal microscopy to take videos at 1 frame every 2 s to enable
manual tracking of mitochondria. We tracked interphase mitochondria within a 125 μm² region of interest (ROI) through actin wave
assembly, disassembly, and post-wave reestablishment of basal mitochondrial network connectivity. As a control, in the same cell we
tracked mitochondria in a separate but equally-sized ROI that was not
sampled by the wave over this time course. Then we recorded the
fraction of original mitochondria that re-fused with distinct neighbors
(Fig. 5C). This fraction was higher for mitochondria recently sampled
by the wave (0.42 vs 0.2 on average), in line with our hypothesis that
hetero-fusion is promoted by this actin wave.

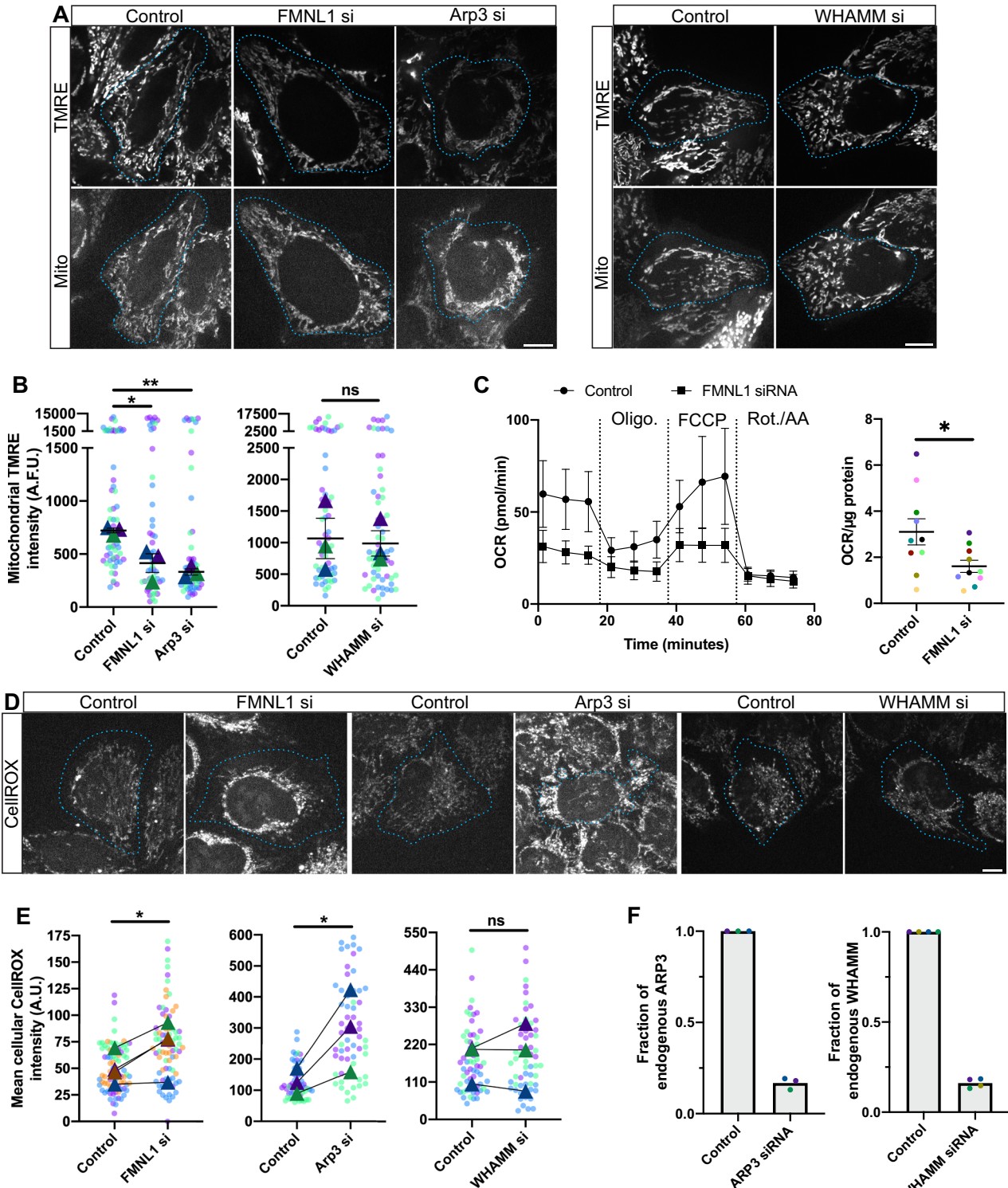

## The interphase actin wave fragments mitochondria in a microtubule-dependent manner

Why does the actin wave induce mitochondrial fragmentation in interphase cells and mitochondrial motility in metaphase cells? One possibility is that the wave is driven by distinct mechanisms during these different stages of the cell cycle. To probe this possibility we reanalyzed data examining the effects of siRNA-mediated depletion of factors known to affect actin polymerization/depolymerization on actin wave size[8]. We separated these previously pooled data into two groups based on whether the cells imaged were in interphase or M phase (Fig S5A). Notably, we found that different cell cycle phases responded to our interventions in the same manner. Since the interventions we examined had similar effects on both the interphase and metaphase waves, and the same holds true for our results from FMNL1 depletion experiments (Fig. 1B, C & Fig S1C, D), we conclude that the machinery building and disassembling the actin wave is the same throughout the cell cycle, and that an independent factor may determine whether the wave fragments mitochondria in interphase or promotes mitochondrial motility in metaphase.

**Fig. 3 | The interphase actin wave maintains mitochondrial homeostasis.**
**A** Representative images of TMRE (mitochondrial membrane potential indicator, incubated at 45 nM for 15 mins) and Mito-sBFP2 in live interphase HeLa cells treated with siRNA to FMNL1, ARP3, or WHAMM. Shown in cyan are outlines of cells as determined by Mito-sBFP2 signal. **B** Average mitochondrial TMRE intensity per cell. Triangles represent medians, bars are means with SEM. Left panel, data analyzed by one-way ANOVA with Dunnett's multiple comparison test. Right panel, data analyzed by a two-sided Student's T test. Note break in Y axis, where top and bottom segments are differentially scaled. $n = 3$ biologically independent experiments.
**C** OCR from FMNL1 knock-down and control cells analyzed using Seahorse Bioanalyzer; example traces and quantitation across multiple independent experiments are shown. In trace, points are means and error bars represent SD. In graph, lines indicate means, error bars represent SEM, and a two-sided Student's T test was

applied. $n = 10$ biological replicates. **D** Representative images of CellROX (cellular ROS indicator, incubated at 1:500 for 30 min) in live interphase cells treated with siRNA to FMNL1, ARP3, or WHAMM. Outlines of cells in cyan determined by Cell-Mask signal. **E** Quantitation of average cellular CellROX intensity, triangles represent means, two-sided ratio paired-T tests were used for statistical analysis. For Control vs. FMNL1, $n = 4$ biologically independent experiments. For other comparisons $n = 3$ biologically independent experiments. **F** Quantitation of western blot experiments examining knock-down efficiencies of ARP3 and WHAMM siRNAs. Bars represent means. $n = 3$ and 4 biological replicates for ARP3si and WHAMMsi comparisons, respectively. **A–F** Scale bars are 10 μm. $\geq 3$ independent experiments. Different colors in graphs represent different biological replicates. Source data are provided as a Source Data file. *, **, and *** indicate p-values of $< 0.05$, $\leq 0.005$, and $\leq 0.0005$ respectively.

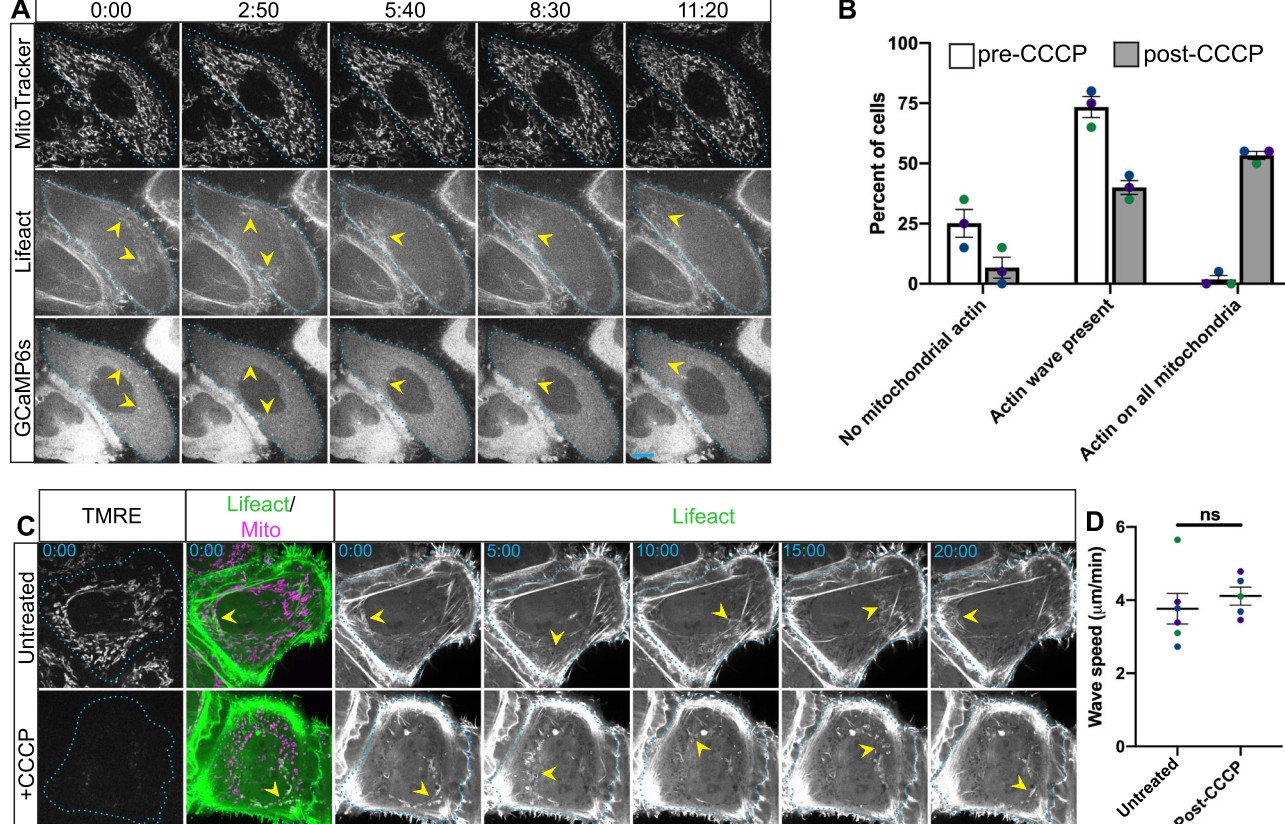

**Fig. 4 | Mitochondrial polarization does not feedback to control the actin wave.**
**A** Timelapse images of live interphase cells expressing Lifeact-miRFP 703 (actin marker) and GCaMP6s (cytosolic calcium indicator) and stained with MitoTracker Red CMXRos. **B** Distribution of cells with different actin/mitochondrial phenotypes in the absence or presence of 20 μM CCCP. Bars represent means with SEM. 3 independent experiments. **C** Timelapse images of live interphase cells expressing Lifeact-GFP, mito-sBFP2, and stained with TMRE (mitochondrial membrane potential indicator) for cells without CCCP addition or cells treated with 20 μM CCCP. Display scaling for the actin and mitochondrial channels was altered

between groups for ease of viewing. **D** Wave speed for untreated and CCCP-treated cells. Means with SEMs are shown; statistical analysis performed using a two-sided Student's T test was employed. $n = 6$ and 5 for untreated and post-CCCP conditions across 3 biological replicates, respectively. Note: different cells observed before and after CCCP addition. **A–D** Dotted cyan lines denote cell boundaries and yellow arrows indicate positions of actin waves. Scale bars are 10 μm. In graphs, different colors represent different biological replicates. Source data are provided as a Source Data file. *, **, and *** indicate p-values of $< 0.05$, $\leq 0.005$, and $\leq 0.0005$ respectively.

We next considered whether differential interactions with the microtubule cytoskeleton might dictate whether mitochondria fragment in response to the actin wave as seen in interphase cells[7], or instead are propelled by actin comet tails as a result of the wave as seen in metaphase cells[8]. In interphase, mitochondria make extensive contacts with the microtubule cytoskeleton, but these contacts are lost in metaphase due to the activity of mitotic kinases that target the microtubule motors that tether mitochondria to microtubules[17]. Also notable, it has been shown that directional mechanical force is

sufficient to induce mitochondrial fission by causing organelle tubulation[41,42]. The force-induced decrease in mitochondrial circumference may facilitate assembly of an oligomeric DRP1 ring to mediate fission[43]. Therefore we asked whether the actin wave fragments mitochondria during interphase by applying force to mitochondria that is resisted by motor-dependent microtubule tethering, thereby inducing mitochondrial deformations such as tubulation leading to thinning in certain areas where fission machinery could operate. To test this hypothesis we stably expressed mito-DsRed2 and

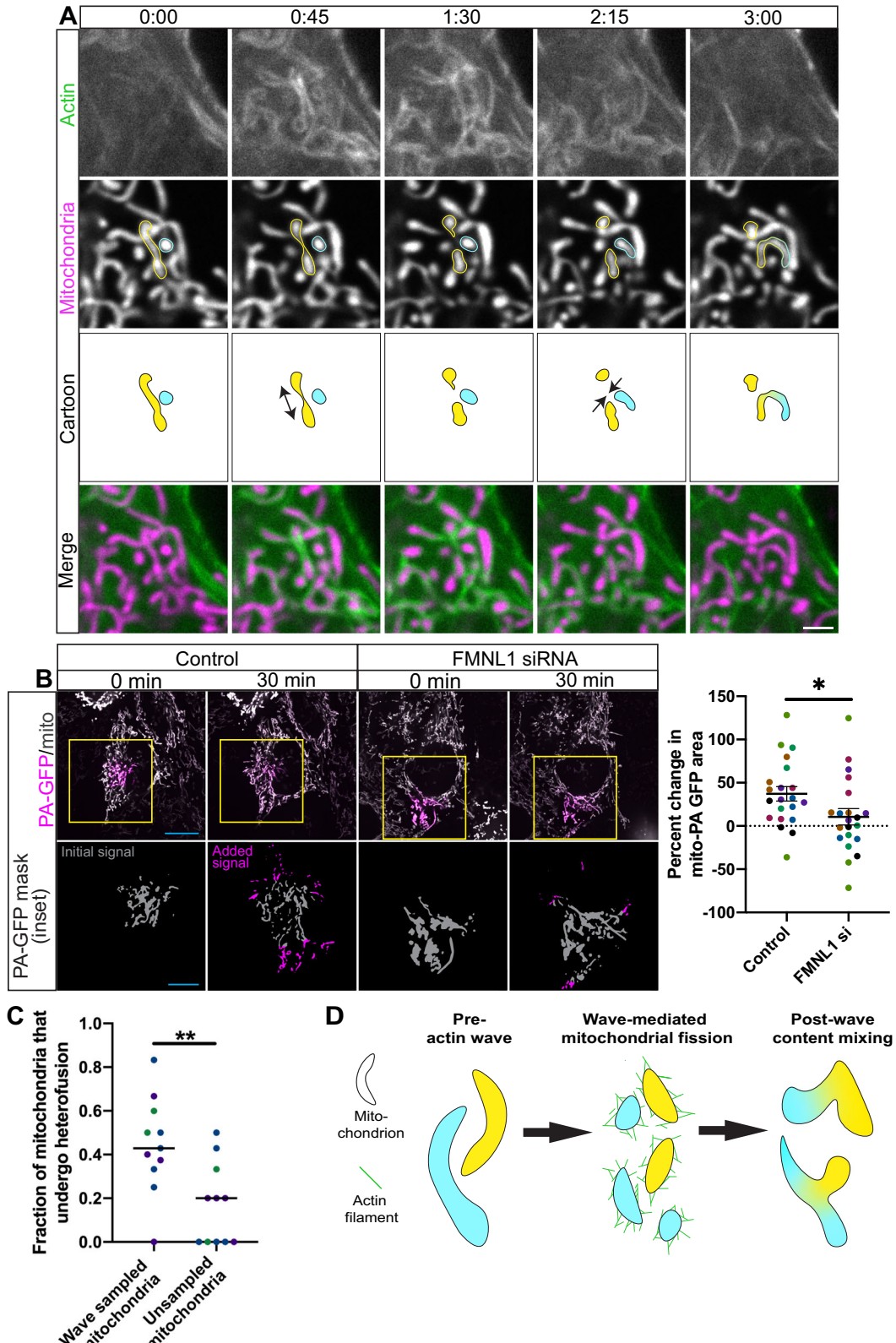

Lifeact-GFP in HeLa cells, then treated these cells with the microtubule-depolymerizing drug nocodazole for one hour at 25 μM, conditions sufficient to depolymerize the interphase microtubule cytoskeleton[44], and conducted live imaging on interphase cells.

In control interphase cells, we observed a robust actin wave leading to local fragmentation of the mitochondrial network, as previously described[7]. In contrast, in nocodazole-treated cells wave-enveloped mitochondria were not fragmented but instead remained as interconnected as mitochondria outside of the actin wave (Fig. 6A). Further, in control interphase HeLa cells the wave did not affect mitochondrial motility, but upon nocodazole treatment wave-enveloped mitochondria were more motile compared to actin-negative mitochondria, as demonstrated by cumulative time projections (Fig. 6B). Indeed, with nocodazole treatment actin wave-

**Fig. 5 | The interphase actin wave promotes mitochondrial hetero-fusion.**
**A** Timelapse images of region of interest in live interphase cell expressing Lifeact-GFP (F-actin marker) and mito-DsRed2. Two distinct mitochondria are outlined in yellow and cyan and depicted as a cartoon. Scale bar is 2 µm. **B** Timelapse images of live interphase cells expressing mito-photoactivatable GFP and mito-DsRed2 after knock down of FMNL1. The total mitochondrial channels were individually scaled for ease of viewing. Approximate segmented PA-GFP signal is shown as a mask. Scale bar is 20 µm for images of whole cells, 10 µm for insets. Accompanying is quantitation for this experiment, where percent change in PA-GFP area was measured. Means and SEMs are depicted and a two-sided Student's T test was performed. $n = 21$ cells across 7 independent experiments. **C** Quantitation from manual tracking of mitochondria recently sampled by actin wave and mitochondria not recently sampled by actin wave. Cells were expressing Lifeact-mScarlet and mito-mEmerald. Bars represent medians and a two-sided Mann-Whitney test was performed for statistical analysis. $n = 11$ cells across 3 independent experiments. **D** Cartoon depicting how the interphase actin wave might promote mitochondrial hetero-fusion. **A–D** In graphs, different colors represent different biological replicates. 3 or more biological replicates. Source data are provided as a Source Data file. *, **, and *** indicate $p$-values of $< 0.05$, $\leq 0.005$, and $\leq 0.0005$ respectively.

enveloped mitochondria had a significantly higher displacement index (see methods for definition): $1.10 \pm 0.18$ vs $0.63 \pm 0.04$, comparing averages with SEMs. In contrast, the displacement index was the same for actin positive and negative mitochondria in control cells.

Strikingly, analysis of movies from nocodazole-treated interphase cells revealed that motile mitochondria within the actin wave typically featured identifiable comet tails (Fig. 6C, D, & Fig S5B). This result was also observed in parallel experiments in COS-7 cells, indicating it is generalizable to a degree (Fig S5C). We could capture details of the dynamics with higher-frame rate imaging (Supplementary Movie 5). The comet tails observed in nocodazole-treated interphase cells closely resemble those previously observed for actin wave-positive mitochondria in dividing cells[8], often featuring two parallel or helically intertwined contrails emanating from the mitochondrial leading edge. In both nocodazole-treated interphase cells and metaphase cells, mitochondria are not associated with microtubules[17]. Notably, intracellular structures that are more free in space, like those dissociated from microtubules, have a higher propensity for forming comet tails driven by surface actin polymerization[45], potentially representing the basis for our novel result. Altogether these data support the model that the actin wave fragments mitochondria in interphase due to their attachments to microtubules, which resist wave force, causing mitochondrial thinning at certain points that could then be accessible to fission machinery (Fig. 6E).

## Discussion

During interphase a wave of F-actin propagates through the mitochondrial network, locally fragmenting these organelles[7]. This actin wave persists into metaphase, but during this stage of the cell cycle the wave ceases to fragment mitochondria and instead promotes mitochondrial motility in a comet tail mechanism[8]. The result of this randomly-directed movement is the spatial mixing of mitochondria within the mother cell, allowing equivalent partitioning of mitochondria between daughter cells. While the function of the metaphase wave has been investigated, a possible function for the interphase actin wave has remained unexplored.

To glean insight into the function of the interphase actin wave, we first sought a more targeted method to inhibit the wave than our previous approaches targeting the ARP2/3 complex, which functions in many cellular contexts. We postulated that a member of the formin-family of actin nucleators/elongators, specifically belonging to the FMNL sub-group, is required for the continued propagation of the wave, and we found that FMNL1 depletion ablated the actin wave. We also investigated FMNL1 is regulated by CDK1 in this context. Such a possibility is feasible even during interphase, as in this cell cycle stage CDK1 activity is low but not absent[46]. During mitosis, CDK1 activity increases, potentially causing the increased propagation speed of the actin wave seen in dividing cells[8]. Consistent with this possibility, we find forced CDK1 activation using adavosertib increases the speed of the actin wave in interphase cells. Further, overexpression of a mutant FMNL1 construct with its CDK1 consensus phosphorylation site changed to a non-phosphorylatable alanine residue (FMNL1 S1031) led to a reduction in the size of the actin wave – supporting the hypothesis that

a CDK1-FMNL1 axis positively regulates the wave. The mutant tested here may exhibit reduced actin polymerizing ability as the altered residue is in the WH2 domain, which is important for actin polymerization[47]. Indeed, overexpression of the analogous mutation in FMNL2 leads to diminished stress-fibers[25], supporting the hypothesis that FMNL1 S1031A is deficient as an F-actin nucleator/elongator. This FMNL1 mutant may inhibit the actin wave because of reduced actin polymerizing ability while still maintaining mitochondrial targeting, thus acting as a dominant negative.

Using FMNL1 knock-down as a tool to ablate the actin wave, we probed for a cellular function in interphase cells. We assayed mitochondrial health after wave ablation by measuring mitochondrial polarization, mitochondrial oxygen consumption, and cellular ROS levels. In each case, our results pointed to a departure from mitochondrial homeostasis, indicative that the cellular function of the interphase actin wave is to maintain mitochondrial health. Though knock-down of FMNL1 also prevents full depolarization-induced F-actin assembly around mitochondria[34], we do not believe this to be a relevant phenomenon for our work, as we found the rate of spontaneous and complete mitochondrial depolarization events to be very low under our cell culture conditions, and of the events we observed only a small fraction featured F-actin accumulation. While our observations indicate that the interphase actin wave maintains mitochondrial health, to our surprise, we find that mitochondrial polarization does not feedback to regulate the wave. Treatment of cells with the mitochondrial uncoupler CCCP induced total depolarization and mitochondrial shape changes, yet was not sufficient to ablate the wave or even to alter its speed of propagation.

We next queried the mechanism by which the interphase actin wave contributes to mitochondrial homeostasis. In our previous work, we identified an instance of actin wave-mediated mitochondrial content mixing during interphase[7]; here we generated quantitative evidence of this phenomenon. So called mitochondrial 'hetero-fusion' is enhanced by the actin wave because the local fragmentation of the mitochondrial network is followed by a return to pre-wave connectivity once the wave departs. The rapid burst of refusion following local F-actin disassembly allows mitochondria to undergo stochastic refusion with neighbors. This mechanism allows for mixing of mitochondrial contents, potentially maintaining mitochondrial homeostasis via complementation, without acutely altering cellular energetics as might occur if all cellular mitochondria were to fragment at once, since mitochondrial size influences mitochondrial distribution, mitophagy, apoptotic signaling, and is at least correlated with mitochondrial metabolic state[48]. Thus the speed and size of the actin wave may be finely tuned – if the wave is too fast/large the mitochondrial network would be constitutively over-fragmented; too slow/small and there would be inadequate mitochondrial content mixing. Several of the proteins that constitute the molecular machinery for the actin wave and dictate its speed and size are either mutated or upregulated in disease, including FMNL1, and deciphering how these perturbations influence the actin wave is now warranted[49,50].

Finally, we hypothesized that the interphase actin wave fragments mitochondria rather than propelling them because the wave

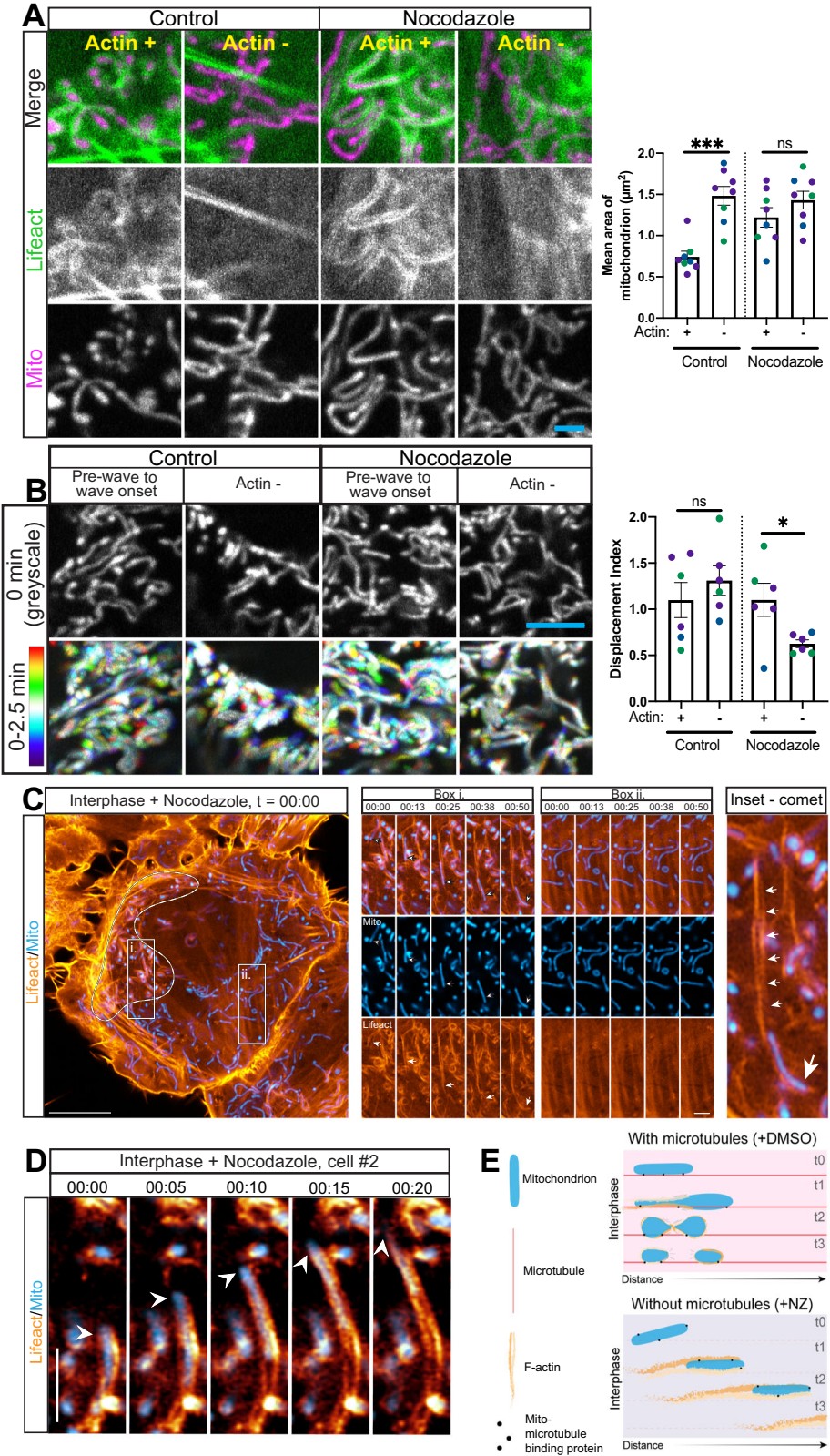

applies force to these organelles which is resisted by microtubule tethering; the application of the resistive force then leads to mitochondrial deformations including tubulation that can lead to localized thinning at certain points, increasing accessibility to fission machinery. Others have shown that stretching of mitochondria can induce mitochondrial fission, and specifically that kinesin/cytoplasmic dynein motility along the microtubule cytoskeleton can mediate this stretching[42]. Further, it has been postulated that mitochondrial deformation may lessen organelle circumference such that an oligomeric DRP1 ring can assemble for fission[43]. To test our hypothesis, we induced microtubule depolymerization with the drug nocodazole. Under these conditions, the interphase actin wave ceased to fragment mitochondria, and instead propelled these organelles, supporting our model.

**Fig. 6 | Mitochondrial tethering to microtubules allows actin wave to fragment mitochondria. A** Representative images of live interphase cells expressing mito-DsRed2 and Lifeact-eGFP after 1 h treatment with DMSO or 25 μM nocodazole, and quantitation of mitochondrial size. $n = 8$ across 3 independent experiments. Scale bar is 2 μm. **B** Cumulative time projections of mitochondria sampled by the actin wave or outside of the actin wave over 2.5 min in either a 1.5–3 h 25 μM nocodazole treated interphase HeLa cell, or a control cell. Probe is Mito-dsRed2. Scale bar is 5 μm. Display scaling between control and nocodazole conditions is not equal for ease of viewing. Also shown is the displacement index of actin-positive and actin negative mitochondria in nocodazole-treated and control interphase cells. $n = 6$ cells over 3 independent experiments. **C** Airyscan. Actin comet tail (Lifeact-eGFP, orange) associated with a mitochondrion (mito-DsRed2, blue) in a 1 h 10 μM nocodazole treated interphase HeLa cell. Scale bar is 10 μm left, 2 μm right. Cell also expressing Halo-Sec61B and labeled with Janelia Fluor 635 (not shown). **D** Another example from a separate cell of actin comet tail (Lifeact-eGFP, orange) associated with a mitochondrion (mito-DsRed2, blue) in a 1 h 10 μM nocodazole treated interphase HeLa cell, obtained by airyscan imaging. **E** Schematic indicating potential effects of actin assembly on mitochondria either bound to or dissociated from microtubule tracks. **A–E** For graphs, differently colored points represent independent experiments. Also, a two-sided Student's t-test was applied for statistical analyses. Further, means are depicted and error bars represent SEM. For each cell observed an actin-positive and -negative measurement was taken. Source data are provided as a Source Data file. *, **, and *** indicate p-values of < 0.05, ≤ 0.005, and ≤ 0.0005 respectively.

Remarkably, wave-enveloped mitochondria in nocodazole-treated interphase cells featured actin comet tails similar to those previously observed in metaphase cells. At this stage of the cell cycle, mitochondria are no longer tethered to microtubules via molecular motors[17], and thus are potentially more free in space. It may be that metaphase cells feature comet tails because actin networks can form comet tails more readily on cellular structures that can be displaced, while this transformation is inhibited on confined objects[45]. Actin polymerization around an object can stochastically produce stronger force on one side, and if the object is non-confined the result is displacement of the object, which is followed by more symmetry breaking of actin-mediated forces with yet further displacement. Iterations of this process lead to highly polarized F-actin in the form of comet tails. Altogether our results are consistent with our hypothesis that the interphase actin wave fragments mitochondria by applying a force that is resisted by microtubule tethering – leading to fission-permissive mitochondrial deformations. Thus, our data shed light on the mechanism by which the interphase wave fragments mitochondria in addition to the cellular function of this process.

In what physiological context are our findings related to the actin wave relevant? While most of our experiments were conducted in HeLa cells, we have been able to reproduce key results in COS-7 cells; further, we have previously observed the wave in primary human keratinocytes[7]. These observations suggest that investigating the role for the actin wave in maintaining mitochondrial health and function in vivo will be an exciting direction for future work.

## Methods
### Reagents
For western blots and immunofluorescence the following primary antibodies were used: TOMM20 (Santa Cruz, sc-17764), FMNL1 (Abcam, ab97456), FMNL2 (Santa Cruz, sc-390298), GFP (abcam, ab1218), GFP (Aves, 1020), phospho-CDK1 substrate consensus sequence (CST, 2325), CDK1 (abcam, ab131450), ARP3 (Proteintech, 13822-1-AP), WHAMM (abcam, ab122572). Coupled to these primary antibodies were the following secondaries: anti-mouse Alexa Fluor 647 (ThermoFisher, A32728), anti-mouse AlexaFluor 405 (ThermoFisher, A31553), anti-Rabbit IRDye 800CW (LI-COR, 926-32213), anti-Rabbit IRDye 680RD (LI-COR, 926-68071) and anti-Mouse IRDye 800CW (LI-COR, 626-32212).

Dyes used were: Phalloidin Alexa Fluor 647 (ThermoFisher Scientific, A22287), Phalloidin Alexa Fluor 488 (ThermoFisher, A22287), MitoTracker Deep Red FM (ThermoFisher Scientific, M22426), Mito-Tracker CMXRos(Cell Signaling, 9082 P), TMRE (Molecular Probes, T-669), CellROX Green (ThermoFisher Scientific, C10444), and Cell Mask Deep Red (ThermoFisher Scientific, C10046), JF SNAP-647 ligand (provided by Luke Lavis, Janelia Farms).

The plasmids used in this study are as follows: Lifeact-EGFP (ref.[7]), Lifeact-mScarlet-N1 (Addgene, 85054), GFP-FMNL1 (subcloned from ref.[24]), GFP-FMNL1 S1031A (subcloned), GFP-FMNL1 S1031E (subcloned), mito-sBFP2-C1 (ref.[7]), GCaMP6s (Addgene, 40753), Lifeact-

miRFP (Addgene, 79993), Mito-DsRed2 (gift from T. Schwarz, Harvard Medical School, Boston), Halo-Sec61B (Addgene, 123285), mito-SNAP (subcloned from Mito-DsRed2 in pSNAPf [New England Biolabs]), and Mito-paGFP (Addgene, 23348).

siRNAs utilized were: Control (ON-TARGETplus, Horizon, D-001810-01-20), FMNL1 (Santa Cruz, sc-62325), Alternative FMNL1 (Horizon pool: CCUCAAACGGGCUGGUGCAUU, GAGCCUGGCCUG UAACUUAUU, and AAGCUGGAUUUCCGAGGCUUU), FMNL2 (Santa Cruz, sc-62327), FMNL3 (Santa Cruz, sc-62329), ARP3 (Santa Cruz, sc-29739), WHAMM (Horizon, L-022415-01), CDK1 (Dharmacon; GUAUAAGGGUAGACACAAAUU).

The following drugs were employed in this study: nocodazole (Sigma, M1404), Ro3305 (Sigma, SML0569), CGP74514a (Enzo, 5472), Adovosertib (MedChemExpress, HY-10993), CK-666 (Sigma, SML0006), CCCP (Sigma, C2759), and Seahorse Mito Stress Test drugs (Agilent, 103015-100).

### Cell culture and transfections
HeLa-M cells were obtained from A. Peden (Cambridge Institute for Medical Research, Cambridge, UK) and COS-7 were from ATCC. HeLa-M cells stably expressing Lifeact-GFP and MitoDsRed2 were previously described in ref.[8]. Cell lines were grown in DMEM (Corning, 10-017-CV) supplemented with 10% FBS and 1% GlutaMAX (Gibco, 35050061), passaged using trypsin, and cultured in an incubator at 37 °C, 5% $CO_2$. Plastic dishes were used for culturing but prior to imaging cells they were plated on #1.5 glass-bottom dishes.

The same protocol was followed for all plasmid transfections. Plasmids were diluted in a volume of OPTI-MEM (Gibco, 31985-070) that was 5% that of the volume of maintenance media and then FuGENE 6 transfection reagent (Promega, E269A) was added. This mix was incubated for 10 min at room temperature and then added to cells 24–48 h prior to imaging, fixation or biochemistry. For cells that were pre-treated with siRNA, prior to plasmid transfection culture media was replaced.

Transfections of siRNA were also all carried out in the same way. In two volumes of OPTI-MEM, each 10% that of the volume of maintenance media, siRNA was diluted in one so that the final concentration on cells would be 40 nM and in the other Lipofectamine RNAiMAX (Invitrogen, 13778-150) was diluted so that the final concentration on cells would be 0.13%. These solutions were mixed, incubated for 10 min at room temperature, and added to cells. All siRNAs were added to cells 48 h prior to start of assays.

### Protein concentration determination and Western blotting
Cells were pelleted, washed with PBS, and RIPA buffer (50 mM Tris-HCl pH 7.4, 150 mM NaCl, 1% Triton-X100, 0.5% deoxycholate, and 0.1% SDS) was added for lysis. In the case of the Seahorse assay, RIPA was added to PBS-washed adhered cells in the 96-well plate. Protease inhibitors were added to RIPA buffer when downstream application was western blotting. Then a BCA assay (ThermoFisher Scientific) was used to measure total protein concentration. Next, unless otherwise

specified equal μg amounts of samples were subjected to SDS-PAGE and transferred onto PVDF membrane (Millipore Immobilon FL 0.45 μm). These were blocked and stained with REVERT Total Protein Dye (LiCor, 926-11011).

Membranes were probed with antibodies at the following concentrations: FMNL1 (1:900), FMNL2 (1:500), abcam GFP (1:1000), Aves GFP (1:5000), phospho-CDK1 substrate consensus sequence (1:500), CDK1 (1:500), ARP3 (1:2000), WHAMM (1:500). These membranes were then incubated overnight at 4 °C, followed by incubation with fluorescent secondary antibodies at 1:20,000 for one hour at room temperature. After washing membranes were then scanned using a Odyssey CLx imaging system and blots were analyzed with Image Studio. To correct for loading unequal μg amounts of samples antibody signal was normalized to total protein stain signal except for in immunoprecipitation experiment since non-equal μg amounts were loaded. See below.

## Immunoprecipitations
In order to pull down GFP-FMNL1 for mass spectrometry, transiently transfected HeLa cells were were scraped into lysis buffer (50 μl/ml NP-40, 5 mM HEPES, 10 mM NaCl, 0.2 mM DTT, 0.001% Brij 35, 0.01 mM EGTA, 0.01 mM MnCl₂, 5% glycerol, pH 7.5, 1X Halt protease and phosphatase inhibitor cocktail from Thermo Scientific #78440). Then sample was centrifuged at 10,000 X g for 10 min. After washing GFP-Trap Magnetic Agarose beads (Proteintech, gtma) with wash buffer (10 mM Tris pH 7.5, 150 mM NaCl, 0.5 mm EDTA, 0.4% Triton-X) and lysis buffer, the supernatant was added to the beads for a one hour incubation at 4 °C with rocking. The beads were then washed three times with wash buffer and then three times with PBS before aspirating all remaining liquid and flash freezing for mass spectrometry processing.

Immunoprecipitation of GFP-FMNL1 for western blot analysis was performed on transiently transfected HeLa cells, which were scraped into lysis buffer above or 0.5% NP-40, 0.15 M NaCl, 10 mM Tris-HCl pH 7.5, 1.7 mM EDTA, 1X Halt protease and phosphatase inhibitor cocktail. The ensuing protocol was then the same as for sample preparation for mass spectrometry, but after wash buffer washes loading buffer was added to beads and they were heated at 95°. For displayed blot (Fig. 2C), initial blot was run and probed with GFP antibody to discern relative amounts of immunoprecipitated protein per sample, then a follow up blot (shown) was run where the amount of each sample loaded was such that pulled-down FMNL1 levels would be equal.

## Mass spectrometry
The immunoprecipitated sample was solubilized and then digested with the iST kit (PreOmics GmbH, Martinsried, Germany) per the manufacturers protocol. The resulting peptides were de-salted and then dried by vacuum centrifugation, followed by reconstitution in 0.1% TFA containing iRT peptides (Biognosys Schlieren, Switzerland). The peptides were analyzed on a QExactive HF mass spectrometer (ThermoFisher Scientific San Jose, CA) coupled to an Ultimate 3000 nano UPLC system and an EasySpray source using data dependent acquisition (DDA). The raw MS files were processed using MaxQuant and visualized in Scaffold (Proteome Software, Portland Oregon).

## qPCR
In order to isolate cDNA, total RNA from cells was collected with TRIzol reagent (ThermoFisher Scientific, 15596018). To further purify bulk RNA, a *Quick*-RNA Miniprep Plus Kit (Zymo Research, R1057T) was utilized, where 50 μl of sample was mixed with 100 μl of RNA Lysis Buffer and 150 μl 100% ethanol. This solution was then transferred to a Zymo-Spin IIICG Column and the protocol prescribed by the kit was then followed except for the DNAase reaction step. Afterwards, this DNAase reaction step was carried out using the following product: Invitrogen, 18068-015. Next, using SuperScript III Reverse Transcriptase (Invitrogen, 18080-093), reverse transcription was done. RNA hydrolysis followed by incubating sampled in 100 mM EDTA and 200 mM NaOH for 15 min at 65 °C and finally DNA was cleaned up with an Oligo Clean & Concentrator kit (Zymo Research, D4060).

For qPCR experiments, PowerUP SYBR Green Master Mix (Thermo Fisher Scientific, A25742) was used with 20 μl reactions. The concentration of both forward and reverse primers was 250 nM. Primers used were as follows: GAPDH fwd (5'-CTGGCTACACTGAGCACC), GAPDH rev (5'-AAGTGGTCGTTGAGGGCAATG), FMNL3 fwd (5'-GGA GCTGTGTATGGCTTCAA), FMNL3 rev (5'-GGTCTGGGTATTTCTCCTT-CAC), FMNL1 fwd (5'-GAAGCCCATCCAGACTAA), FMNL1 rev (5'-GT CTTGAACTGTTCCTCAAA). The PCR program for all primer sets was 95 °C for 10 min, and 40 cycles of 95 °C for 15 s then 55.5 °C for 30 s then 60 °C for 30 s. The following amounts of template were used: 1 ng for GAPDH, 10 ng for FMNL3, 120 ng for FMNL1. Percent knock-down was calculated using the ΔΔCt methods.

## Seahorse
To measure mitochondrial oxygen consumption rate with a Seahorse assay first HeLa-M cells were seeded into a Seahorse microplate (Agilent, 103794-100) at 2000 cells per well, with 5 technical replicates for each condition. 24 h later cells were treated with siRNA. After 48 h (with a media change at 24 h post-siRNA), a mito stress test was run. First cells were changed to assay media of 25 mM glucose, 1 mM pyruvate, and 2 mM glutamine (Agilent, 103680-100). Then the assay was run using a Seahorse XF96 Analyzer, with oligomycin at 1 μM, FCCP at 1 μM, and Rotenone/Antimycin A at 0.51 μM. The assay protocol was as follows: 3 min mix, 3 min measure, no wait, and 3 cycles. For analysis, values from technical replicates were averaged.

## Immunofluorescence
Cell were fixed by addition of 4% paraformaldehyde immediately after removal of maintenance media for a 10 min incubation at 37 °C. After 3 PBS washes cells were permeabilized for 5 min with 0.2% Triton-X100. Following this step, cells were blocked in buffer containing 0.5% BSA, 2.5% normal goat serum, and 2.5% normal donkey serum at 37 °C for 30 min. With another PBS wash primary antibody was diluted in 0.5% BSA and added to cells for overnight incubation at 4 °C. TOMM20 primary antibody was added at a concentration of 1:40. The next day cells were washed 3 times with PBS and secondary antibody and phalloidin were diluted to 1:300 and 1:40 in 0.5% BSA in PBS, respectively, and added for a one hour room temperature incubation. After another round of PBS washing, cells were covered with Slow Fade Diamond Mounting Media (ThermoFisher Scientific, S36972) for imaging.

## Confocal microscopy
Most confocal microscopy was conducted with a spinning-disk system (UltraView VoX; Perkin Elmer). This system was mounted on a Nikon Eclipse Ti Microscope equipped with an Apochromat 100 × 1.49 numerical aperture oil immersion objective (Nikon) as well as a charge-coupled device camera (C9100; Hamamatsu Photonics) or a CMOS camera (Orca-fusion, Hamamatsu Photonics). Only imaging in Fig. 6C, D & Fig S5B was carried out with a Zeiss LSM 880 equipped with an Airyscan module and plan apochromat 63x/1.4 NA DIC M27 oil objective. Within each experiment all microscope acquisition parameters were kept constant and the same number of cells were imaged per group. The latter was true except for Fig. 2D. Neither statement applied to Fig. 2E or Fig S5A, though here only small alterations to microscope acquisition parameters were made. n = at least 10 for each wave size experiment unless otherwise specified in figure. In many cases prior to imaging live cells MitoTracker was added to maintenance media at 50-25 nM for 10 min. For the experiment in Fig S3C SNAP labeling was carried out first. After one wash cells were put in

imaging media, composed of 10% FBS and 1% GlutaMAX (Gibco, 35050061) in Leibovitz's L-15 medium (Gibco, 11415-064), and set for ~1 h at 37 °C in the microscope chamber. In experiments where mitochondria and actin markers were stably expressed, cells were imaged in their regular culture media. For all experiments, cells to image were chosen randomly with respect to the measured or noted phenotype. In the photo-activatable GFP experiment, cells with mitochondria of equivalent connectivity (as determined qualitatively) were specifically chosen. In actin wave size experiments, z-plane where wave was largest was acquired.

## Image analysis

To determine mitochondrial size, regions of interest (ROIs) were chosen based on highest mitochondrial fragmentation signature in the actin channel. Then, mitochondria were manually segmented and the Fiji 'analyze particles' function was used to determine the average size of a mitochondrion in the ROIs. Analysis was conducted in a blinded manner. Cumulative time projections of mitochondrial signal (Fig. 6B) were manually thresholded in Fiji in a blinded manner. Displacement index was defined as the area mitochondria encompass at the end of a cumulative time projection divided by the area they encompass at the start and subtracted by 1. For wave size analyses, the image observer was blinded to file names and then manually drew regions of interest around clustered, F-actin-positive mitochondria in the program Fiji (Fig S1A). Blinding was true except for in Fig. 2H, Figures S1C and S6A. In determining the maximum wave size in control and FMNL1-depleted cells, the movie frames that featured the actin wave at its largest as determined qualitatively was chosen for analysis. All tracking was done manually, and in a blinded manner except for Fig. 4D. The wave centroid (Fig S1A) was used for tracking its propagation and calculating its speed. In measuring background subtracted mean mitochondrial TMRE signal each cell had its mitochondria segmented using intensity thresholding of mito-BFP signal - this was used as a mask for the reported measurement. For the determination of cellular ROS levels cells were outlined using CellMask signal as a guide, and background subtracted mean CellROX signal over that area was recorded. Occasional negative values owing to low basal CellROX signal were converted to zeros. In the photoactivation experiments, mitochondria labeled upon photoconversion were manually segmented using intensity thresholding after blinding. The sum of the areas of objects within cells was recorded at the beginning and end of time courses. Mito-DsRed2 signal was not used in quantitation.

## Reporting summary

Further information on research design is available in the Nature Portfolio Reporting Summary linked to this article.

## Data availability

Mass spectrometry data have been deposited to the ProteomeXchange Consortium via the PRIDE partner repository with the dataset identified PXD050085. The raw imaging data are available under restricted access for reason of large file sizes, access can be obtained by requesting from the corresponding author. Source data are provided with this paper.

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

## Acknowledgements

We thank Mariko Tokito for assistance in cloning. Further, we thank the CHOP-Penn Proteomics Core Facility (RRID:SCR_023099) for help with mass spectrometry experiments. Finally we acknowledge our fundings sources - this work was supported by National Institutes of Health grants R35 GM126950 to E.L.F.H. and RM1 GM136511 to E.L.F.H. and E.M.O.

## Author contributions

S.M.C. and E.L.F.H. conceived the study and wrote the manuscript. S.M.C., A.S.M., E.M.O. and E.L.F.H. designed experiments. S.M.C., A.S.M., C.P.T. and C.F.T. performed experiments. S.M.C. and A.S.M. performed data analysis. A.S.M., C.P.T., C.F.T. and E.M.O. edited the manuscript.

## Competing interests

The authors declare no competing interests.
