## [Peer Review File · Nature Communications]

An interphase actin wave promotes mitochondrial content mixing and organelle homeostasisREVIEWER COMMENTS

Reviewer #1 (Remarks to the Author):

In this manuscript, Coscia et al. explore the phenomenon of mitochondrial-associated actin waves that cycle rotationally through the cytoplasm of interphase cells. Previously work from the lab characterized these waves as associated with mitochondrial fission and subsequent re-fusion in interphase cells, and with actin-based mitochondrial motility in mitotic cells. Here, the authors utilize a candidate approach to identify the formin FMNL1 as required for actin waves. They identify a putative CDK1 phosphorylation site on FMNL1 and provide evidence consistent with phosphorylation as a potential regulatory mechanism. The authors then explore the physiological consequences of FMNL1 knockdown, identifying decreased mitochondrial membrane potential, increased cellular ROS, and decreased cellular ATP, which they attribute to a loss of actin waves. The authors additionally provide quantitative evidence that mitochondrial fusion is disrupted by acute inhibition of ARP2/3. These data collectively support the model that actin waves promote mitochondrial health by promoting context mixing of mitochondria. Finally, the authors explore the relationship between mitochondrial interactions with microtubules and actin, finding that microtubule depolymerization leads to an increase in mitochondrial trafficking along actin akin to what is observed in mitotic cells.

The manuscript is interesting and provides novel mechanistic and functional insights into the phenomenon of mitochondrial-associated actin waves. However, a concern is that while FMNL1 knockdown clearly affects actin waves, it is not evident whether the mitochondrial functional defects associated with FMNL1 depletion are specific to loss of waves of actin, all actin-mitochondrial association, or potentially to other pleiotropic effects. Additionally, key observations in the manuscript are underexplored, such as the implied defects in mitochondrial content mixing due to loss of actin waves and the putative phospho-regulation of FMNL1.

Specific points:

1. The authors should clarify if FMNL1 depletion generally affects actin association with mitochondria or specifically prevents cycling actin. The quantification of actin association in fixed cells in Fig. 1B does not necessarily report on actin cycling unless the only actin associated with mitochondria is cycling. Does actin only interact with mitochondria as part of a wave? The authors should also report the percent of cells with actin/mitochondrial association in control and siFMNL1 (Fig. 1C) and the percent of live cells visualized with static mitochondrial-associated actin and cycling actin (Fig. 1E).
2. The authors conclude that reduced mitochondrial fusion after treatment with CK-666 suggests actin waves promote mitochondrial mixing to promote complementation. However, given that both actin and actin waves are associated with mitochondrial fission, does the reduced fusion instead reflect that mitochondria are no longer dividing (and thus re-fusing) in response to actin waves? To test if actin waves promotes true complementation, would it be possible to examine this in a hybrid fusion of

mtDNA-positive and rho0 cell lines, heteroplasmic cells, or another tool that locally disrupts mitochondrial function. This may be a technical challenge, but otherwise the data shown merely recapitulate the authors' previously published observation in a quantitative manner.

3. Mitochondrial fusion is assayed after acute ARP2/3 inhibition. Is mitochondrial fusion or morphology affected by FMNL1 depletion?

4. While the data presented are consistent with the model that CDK1 acts on FMNL1, this is not directly shown and the conclusion that "CDK1-FMNL1 axis positively regulates the mitochondrial actin wave" is not supported. Such a conclusion requires in vitro evidence that CDK1 directly phosphorylates FMNL1. Alternatively the authors could test by mass spectrometry if FMNL1 is phosphorylated at the S1013 site in a CDK1-dependent manner. Without such data, it remains possible that the S1013A mutation interferes with endogenous FMNL1 function independently of phosphorylation status. The authors should also show that the GFP-FMNL1(S1013A) expresses as well as the wild type version of the protein by Western blot and that the fusion protein migrates at the correct size and is not cleaved or degraded.

5. What is the effect of phospho-mimetic FMNL1 S1013E? Does it increase the frequency or speed of actin waves on mitochondria? Does it cause FMNL1 to more stably associate in proximity to mitochondria? Conversely, does the S1013A mutation in FMNL1 affect wave frequency or general actin association with mitochondria? The inset in Figure 2D for S1013A shows an area with diminished actin, but there are clearly other regions of the cell where actin does appear to concentrate around mitochondria.

6. As altered membrane potential does not always reflect a defect in mitochondrial function, the authors should examine if FMNL1 KD cells have respiratory defects as measured by oxygen consumption or cell growth defects in media that requires respiration (ie galactose).

7. Images of protein staining used to quantify Western blots should be shown in Figure S1A or a loading control should be shown. Western blots should also be shown to validate other knockdowns (ARP3 and WHAMM).

8. While the FMNL3 knockdown is not an essential finding for publication, the knockdown must be validated at least by qPCR in order to draw meaningful conclusions.

9. Figure 2 and Figure S1 are labeled for x-TOMM20 rather than anti-TOMM20. Is this a typo?

Reviewer #2 (Remarks to the Author):

Review of "An interphase actin wave promotes mitochondrial content mixing and organelle homeostasis" by Coscia et al.

Endoplasmic waves of F-actin that associate with mitochondria are a recently-discovered, and fascinating phenomenon. Their role in m-phase is clear: as this lab demonstrated, the waves help disperse the

mitochondria ensuring that the nascent daughter cells receive roughly equal proportions of roughly equivalent (in terms of health) mitochondria. In the current study, the authors have sought to understand the role of interphase mitochondrial waves. To address this point, the authors identified a key regulator of mitochondrial associated waves, the formin FMNL1. They demonstrate that knockdown of this protein suppresses the waves and, strikingly, causes a reduction in the mitochondrial membrane potential. They explain this result as arising from a reduction in mitochondrial fission and a resultant reduction of mitochondrial “complementation”, in which fusion with healthy neighbors restores the health of sickly mitochondria. They also find that the fission depends on mitochondrial anchoring to microtubules and report that FMNL1 is positively regulated by cyclin-dependent kinase 1, the so-called “master regulator of m-phase”.

This is a great story. However, some of the points rest on experiments that are not as solid they could be. These are discussed below:

1. The Cdk1 part of this work is hard to understand. In interphase cells, Cdk1 activity should be low or, if one believes the dogma, non-existent, because it has not yet undergone activation via cyclin B interaction simply because the cyclin B isn't available until late G2. So why do the Cdk1 inhibitors repress waves? One possibility is that they are actually targeting other Cdks, such as Cdk2 or 4/6 which are active in G2 and G1/S respectively. The kinase inhibitors used are more potent at very low nanomolar concentrations for Cdk1 than the other Cdks, but at ~100-200 nM, they also inhibit other Cdks. The actual concentration used was not mentioned (or if it was, I couldn't find it) and in any case, the external concentration often ends up being lower than the internal concentration for many drugs due to trapping via target binding. The use of the wee1 inhibitor, which should stimulate Cdk1, is a nice complementary approach, except that rather than showing it has the opposite effect as the Cdk1 inhibitors—reduction of wave size—the authors simply report that it increases wave speed (see also below). If the authors really wish to demonstrate that FMNL1 is downstream of Cdk1, they should deplete it or use an alternative approach to limit its activity such as over expression of a protein inhibitor. However, they could also simply change their conclusion to FMNL1 being downstream of Cdks in general.

2. It is not clear why the authors jump back and forth from wave area to wave speed both for the Cdk1 experiments and the CCCP experiment vs the other experiments. In principle, these report on different features of wave function rather than being interchangeable. For the sake of consistency, and since most of the measurements are based on area, such measurements should be done for both the wee1 inhibition and the CCCP treatment.

3. The heterotypic fusion experiments should really be done with the FMNL1 knockdown rather than the CK-666, as the former is much more specific (as the authors acknowledge in the Results section). Unless, of course, there is some technical reason this is impossible.

4. The measurements for the heterotypic fusion experiments were difficult to understand. The wave sample vs. unsampled experiments would be reasonable if the authors had measured the ratio of the general mitochondrial marker to the PA-GFP. It's possible they did this but, if so, it isn't clear from the description. The with and without CK-666 experiments seem to be subject to the concern that having actin around will naturally make it more likely that the mitochondria will undergo displacement. Thus, measuring a change in the area of the PA-GFP signal would not distinguish between the mitochondria moving more due to the actin wave and actual heterofusion. Again, it seems more appropriate to directly track fusion by monitoring the extent of mixing of the generic mitochondrial marker with the PA-GFP signal.

Reviewer #3 (Remarks to the Author):

Coscia et al. report the roles of FMNL1 and CDK1 in regulating a wave of actin polymerization that cycles around the cell, within HeLa cells. They further investigate how the same wave has different repercussions for mitochondrial dynamics (fission vs comets) depending on the context.

Overall, the claims are well supported by the data, and the work is original and represents a significant advance in the field of mitochondrial dynamics. However, there are a few flaws in the reporting and analysis, which require revision.

1. The actin wave in Fig. 1A is difficult to spot: perhaps a differential or ratiometric analysis between subsequent images would better highlight where the dynamics are? Also, the actin "wave" in Fig. 2C (control) does not circulate as the authors suggest is the typical pattern. Please comment on this – is this non-circular trajectory common?

2. Some of the image analysis methodologies should be more clearly described:

a) How was TMRE quantified? If absolute intensities were used, they can be misleading due to differential uptake: this is mitigated by normalizing against a different probe.

b) How the quantitative analysis of actin wave (area determination, speed determination) is done is unclear, a section on this with a schematic figure would help, and code should be made available.

c) Analysis of photoconversion expansion: The authors use Ilastik and difference in area of thresholded objects. The spreading is low during the timeframe used, and using intensity values instead of thresholding would be more sensitive. For example, analyzing the variance of the intensity distribution at different timepoints. I also found the image display (green on grey) confusing, as grey looks like a dim green.

d) Statistics to revisit:

- i. Fig 2B: TMRE uptake is variable, as seen by the two populations (low and high). A justified exclusion of the statistical outliers from the analysis would make sense, or normalising (as in previous point).
- ii. SEM is used instead of SD, this should be justified.
- iii. Statistical tests used (one-way ANOVA and student's t test) assume normal distribution of data, which in this case it doesn't seem to be. Normality tests and Mann-Whitney and Kruskal-Wallis one-way ANOVA tests should be performed.
- iv. Some figures need more cells to support the statistics (e.g. Fig 5C)
- v. Videos would benefit from slower playback for ease of seeing the wave.

3. Fig. 6A is missing the quantification of size (+/-actin) for untreated cells. Figure 6B should color code the time, to allow the reader to better observe the dynamics. Figure S3B: This would be the corresponding plot to Fig. 6, but it lacks mito size quantification for (+/-actin) shown in 6B.

4. The study is performed on a single cell line, HeLa. The authors should comment in the discussion about how universal they expect their findings to be. They have previously published limited data on other cell types (NHEK, HaCaT, HEK 293T, COS-7, and iPS). However, the wave in many of those was not as striking and, in some cases, barely discernible to this reviewer. It does not diminish the interest of the work, but other researchers in the field would benefit from understanding the expected limitations versus universality of the findings.

Minor points:

5. Lines 444-447: This sentence doesn't make sense to me.
6. Line 415: The link between mitochondrial size and metabolic state is correlative and not causal, to my knowledge.
7. Figure 2E: Wording of legend is confusing.
8. Figure 4A: Typo in Lifeact label

Response to Review, *Coscia et al.*

We thank the reviewers for their thoughtful comments and advice. We have taken care to address each comment, with details below, and agree that these revisions and additional new data significantly strengthen our work. Thus we hope that our revised manuscript is now acceptable for publication in *Nature Communications*.

Reviewer 1

The quantification of actin association in fixed cells in Fig. 1B does not necessarily report on actin cycling unless the only actin associated with mitochondria is cycling. Does actin only interact with mitochondria as part of a wave?

Actin-mitochondria interactions beyond the wave have been described (for example in Fung et al., *Current Biology* 2022 & Quintero et. al., *Current Biology* 2009). However, unlike the actin wave, none of these interactions are apparent in single-timepoint fluorescence micrographs of unperturbed cells. Thus, the wave size measurements featured in this manuscript indeed reflects this phenomenon. We now clarify this point in the text related to Figure 1.

The authors should clarify if FMNL1 depletion generally affects actin association with mitochondria or specifically prevents cycling actin.

FMNL1 depletion has been shown to interrupt assembly of F-actin on depolarized mitochondria (Fung et. al., *Current Biology* 2022) but this phenomenon is barely seen in cells untreated with mitochondrial toxin, as we demonstrate here (Fig S3C). Possible effects of FMNL1 knock-down on other mitochondria-actin interactions unrelated to the wave remain to be investigated, but our analysis suggests that the primary effect of FMNL1 depletion in unperturbed cells is the inhibition of the mitochondrially-associated actin wave. The text related to Figure 3 has been amended to reflect this reasoning.

The authors should also report the percent of cells with actin/mitochondrial association in control and siFMNL1 (Fig. 1C) and the percent of live cells visualized with static mitochondrial-associated actin and cycling actin (Fig. 1E).

(Note: Fig1E is now Fig S1I) We agree that reporting the percentage of fixed cells that feature an actin wave after transfection with either a control oligo or FMNL1 siRNA more clearly describes the phenotype observed. Thus, we have incorporated these data into our revised manuscript (Fig 1D), finding that FMNL1 knock-down decreases this percentage, consistent with the hypothesis that this protein is a negative regulator of the wave.

While we understand the rationale behind asking about the percent of live cells with static and dynamic actin assemblies on mitochondria, we typically do not observe static mitochondrially-associated actin, only the dynamic association of actin with mitochondria that we describe here and in our previous related studies (Moore et al., *Nature Communications* 2016 & Moore et al., *Nature* 2021). Therefore the second measurement suggested here cannot be taken.

The authors conclude that reduced mitochondrial fusion after treatment with CK-666 suggests actin waves promote mitochondrial mixing to promote complementation. However, given that both actin and actin waves are associated with mitochondrial fission, does the reduced fusion instead reflect that mitochondria are no longer dividing (and thus re-fusing) in response to actin waves?

Indeed, our belief is that inhibition of the ARP2/3 complex via CK-666 or siRNA-mediated depletion of FMNL1 leads to reduced mitochondrial fusion because these treatments eliminate the effect of the actin wave in driving mitochondrial fission, thus allowing for subsequent re-fusion of distinct organelles. To clarify, these inventions block wave-mediated fission because

they ablate the wave itself. In our revised manuscript we have amended the text related to Figure 5 to better reflect our reasoning.

To test if actin waves promotes true complementation, would it be possible to examine this in a cybrid fusion of mtDNA-positive and rho0 cell lines, heteroplasmic cells, or another tool that locally disrupts mitochondrial function. This may be a technical challenge, but otherwise the data shown merely recapitulate the authors' previously published observation in a quantitative manner.

We agree that suggested experiments with the outlined approach might provide additional data supporting the contribution of the actin wave to mitochondrial complementation but, as noted by the referee, technical challenges prevent us from taking this path. Instead, in our revised manuscript we provide quantitative evidence via two distinct experiments (Fig 5B&C) for the hypothesis that the actin wave promotes mitochondrial content mixing. This hypothesis was initially proposed based on occasional observations of re-fusion of distinct mitochondria after the wave (Moore et. al., Nature Communications 2016); here we provide a quantitative analysis of the phenomenon for the first time. Further, the effects of complementation were previously unexplored, and here we assess this aspect in several ways including a TMRE loading assay and a Seahorse mitochondrial stress test, with the data indicating that indeed the wave promotes complementation and maintains mitochondrial health (Fig 3 & S3). Therefore we believe this new work constitutes an important advance over our previous studies.

Mitochondrial fusion is assayed after acute ARP2/3 inhibition. Is mitochondrial fusion or morphology affected by FMNL1 depletion?

We have now investigated mitochondrial morphology following treatment of cells with FMNL1-targeting siRNA and detect no change from baseline (Fig S4). Additionally, for our revised manuscript we assayed mitochondrial fusion after FMNL1 knock-down using our PA-GFP experimental paradigm (Fig 5B) and find fusion is diminished. Collectively, these new results further support our hypothesis that the actin wave plays a role in facilitating mitochondrial content mixing.

While the data presented are consistent with the model that CDK1 acts on FMNL1, this is not directly shown and the conclusion that "CKD1-FMNL1 axis positively regulates the mitochondrial actin wave" is not supported. Such a conclusion requires in vitro evidence that CDK1 directly phosphorylates FMNL1. Alternatively the authors could test by mass spectrometry if FMNL1 is phosphorylated at the S1013 site in a CDK1-dependent manner. Without such data, it remains possible that the S1013A mutation interferes with endogenous FMNL1 function independently of phosphorylation status.

We thank the reviewer for this insightful comment and have addressed this point directly in our revised manuscript. We performed mass spectrometry as suggested, and can now demonstrate that S1031 in FMNL1 can be phosphorylated under basal conditions (Fig 2B). Further, we found that upon pharmacological activation of CDK1, FMNL1 demonstrated enhanced phosphorylation, as determined by reactivity with an antibody which recognizes phosphorylated CDK1 substrate consensus sequences on western blots (Fig 2C). These data, in conjunction with the previous literature describing FMNL1 S1031 as part of the CDK1 substrate proteome, strongly suggest that indeed CDK1 phosphorylates FMNL1 at S1031.

The authors should also show that the GFP-FMNL1(S1013A) expresses as well as the wild type version of the protein by Western blot and that the fusion protein migrates at the correct size and is not cleaved or degraded.

We agree that these points are important to establish. First, upon discovery of several extraneous C-terminal amino acid residues in the previously published GFP-FMNL1 construct

we used for our initial studies, we corrected this construct and re-did the relevant experiment. Importantly, our previous results were repeated with this corrected construct (Fig 2K & L). We now include a graph describing these data, and an additional plot describing observations with a S1031E construct (Fig 2O & P), where we show the mean cellular GFP-FMNL1 intensities for all the WT and mutant expressors we analyzed (Fig 2N & R); on average there is no difference. To further strengthen the manuscript, we now include a western blot (Fig 2M & Q) indicating that WT and mutant constructs of GFP-FMNL1 migrate at the correct size and not are meaningfully degraded.

What is the effect of phospho-mimetic FMNL1 S1013E? Does it increase the frequency or speed of actin waves on mitochondria? Does it cause FMNL1 to more stably associate in proximity to mitochondria? Conversely, does the S1013A mutation in FMNL1 affect wave frequency or general actin association with mitochondria?

We have now assessed the effect of GFP-FMNL1 S1031E overexpression on the actin wave. Specifically, we find this intervention increases wave size and does not alter either wave frequency or speed (Fig 2O & P, Fig S2E), consistent with a role for phosphorylation at S1031 positively regulating the actin wave. Further supporting this idea, we also now assess wave frequency after overexpression of the S1031A mutant and find this metric to be decreased (Fig S2D). While we now include imaging data showing the localization of FMNL1 to actin wave-enveloped mitochondria (Fig 1F), due to the high cytosolic pool of FMNL1 we were not able to quantitatively compare the extent of recruitment of the wild type and phospho-mimetic constructs with sufficient accuracy to include these data in the revised manuscript.

The inset in Figure 2D for S1013A shows an area with diminished actin, but there are clearly other regions of the cell where actin does appear to concentrate around mitochondria.

(Now refers to Fig 2K) We thank the reviewer for their careful assessment of our manuscript. We believe the other regions referred to feature dense sub-cortical F-actin but not peri-mitochondrial actin as seen in the other conditions. Regardless, upon discovering several extraneous C-terminal amino acids in our FMNL1 constructs we made the appropriate correction of the plasmid and re-did this experiment and so now present new representative images (Fig 2K). Importantly, our results with the corrected plasmid fully confirmed our previous observations.

As altered membrane potential does not always reflect a defect in mitochondrial function, the authors should examine if FMNL1 KD cells have respiratory defects as measured by oxygen consumption or cell growth defects in media that requires respiration (ie galactose).

We agree that this is an important point to address. In our revised manuscript we now include data assessing the mitochondrial oxygen consumption rate in FMNL1-depleted cells using a Seahorse Bioanalyzer (Fig 3C) and find this metric decreased, in support of our hypothesis that FMNL1, and by extension the actin wave, maintains mitochondrial homeostasis.

Images of protein staining used to quantify Western blots should be shown in Figure S1A or a loading control should be shown.

(Refers to Fig S1B & H in our revision) Throughout the manuscript we now include total protein stain blots to accompany any blots probed with antibody and associated quantitation. Also, in revising our manuscript we noticed that we accidentally mislabeled our quantitation of FMNL2 siRNA knock-down efficiency as representing FMNL1 siRNA #2 knock-down efficiency and vice versa. We have now corrected this error and note that the two siRNAs similarly, robustly depleted their targets, so no conclusions were affected.

Western blots should also be shown to validate other knockdowns (ARP3 and WHAMM).

In our revised manuscript we now include western blots demonstrating that our ARP3 and WHAMM siRNAs strongly deplete these proteins (Fig 3F & S3B).

While the FMNL3 knockdown is not an essential finding for publication, the knockdown must be validated at least by qPCR in order to draw meaningful conclusions.

We now report that, as measured by qPCR, our FMNL3 siRNA meaningfully depletes this protein (Fig 1E).

Figure 2 and Figure S1 are labeled for x-TOMM20 rather than anti-TOMM20. Is this a typo?
Here we mean anti-TOMM20 and have edited the relevant labels for clarity.

Reviewer 2

The Cdk1 part of this work is hard to understand. In interphase cells, Cdk1 activity should be low or, if one believes the dogma, non-existent, because it has not yet undergone activation via cyclin B interaction simply because the cyclin B isn't available until late G2...

We believe that CDK1 activity is best characterized as low but not non-existent during interphase. Indeed, various studies have provided evidence for Cdk1 phosphorylation of substrates outside of M phase (for example, Jones et. al., JCB 2018 demonstrate phosphorylation of FMNL2 by CDK1 during interphase). In our revised manuscript we now make this point more clearly in the discussion section.

...So why do the Cdk1 inhibitors repress waves? One possibility is that they are actually targeting other Cdks, such as Cdk2 or 4/6 which are activity in G2 and G1/S respectively. The kinase inhibitors used are more potent at very low nanomolar concentrations for Cdk1 than the other Cdks, but at ~100-200 nM, they also inhibit other Cdks. The actual concentration used was not mentioned (or if it was, I couldn't find it) and in any case, the external concentration often ends up being lower than the internal concentration for many drugs due to trapping via target binding... If the authors really wish to demonstrate that FMNL1 is downstream of Cdk1, they should deplete it or use an alternative approach to limit its activity such as over expression of a protein inhibitor. However, they could also simply change their conclusion to FMNL1 being downstream of Cdks in general.

We appreciate the reviewer's insight from the literature and agree it is important to be clear about the relationship between the actin wave, FMNL1, and Cdk1. Thus, in our revised manuscript we assess the effect of CDK1 knock-down on wave size (Fig 2F,I & J; Fig S2C), and find that this intervention causes a decrease, much like the CDK1 inhibitors we employed. We thus conclude that CDK1 is indeed a positive regulator of the wave.

...The use of the wee1 inhibitor, which should stimulate Cdk1, is a nice complementary approach, except that rather than showing it has the opposite effect as the Cdk1 inhibitors—reduction of wave size—the authors simply report that it increases wave speed.

In our revised manuscript we compare actin wave size/frequency for control cells and cells treated with the Wee1 inhibitor (Fig S2A), finding no significant difference. It may be that past a certain threshold of Cdk1 activity, the effect of the kinase is to regulate wave speed. Fitting this model, when cells enter M phase and Cdk1 activity rises, it is the speed of the wave that increases (Moore et al., Nature 2021). We now explore this possibility in the discussion section.

It is not clear why the authors jump back and forth from wave area to wave speed both for the Cdk1 experiments and the CCCP experiment vs the other experiments. In principle, these report on different features of wave function rather than being interchangeable. For the sake of consistency, and since most of the measurements are based on area, such measurements should be done for both the wee1 inhibition and the CCCP treatment.

As described above, we now include an analysis of actin wave size after Wee1 inhibition (Fig S2A). Further, wave speed is now reported whenever possible (Fig S2E). This measurement could not be taken for interventions that downregulate the size/frequency of the wave, as this makes wave tracking difficult. We are unable to measure wave size and obtain meaningful information after CCCP treatment because this treatment results in some cells in which all mitochondria are F-actin positive (Fig 4B) due to a transient depolarization-induced actin assembly that others have described (Fung et. al., Current Biology 2023).

The heterotypic fusion experiments should really be done with the FMNL1 knockdown rather than the CK-666, as the former is much more specific (as the authors acknowledge in the Results section). Unless, of course, there is some technical reason this is impossible. We agree about the value in conducting this experiment using FMNL1 knockdown, and now include these data in our revised manuscript (Fig 5B). We find that PA-GFP spread, and so mitochondrial fusion, is decreased in cells lacking FMNL1 just as is the case with CK-666.

The measurements for the heterotypic fusion experiments were difficult to understand. The wave sample vs. unsampled experiments would be reasonable if the authors had measured the ratio of the general mitochondrial marker to the PA-GFP. It's possible they did this but, if so, it isn't clear from the description. The with and without CK-666 experiments seem to subject to the concern that having actin around will naturally make it more likely that the mitochondria will undergo displacement. Thus, measuring a change in the area of the PA-GFP signal would not distinguish between the mitochondria moving more due to the actin wave and actual heterofusion. Again, it seems more appropriate to directly track fusion by monitoring the extent of mixing of the generic mitochondrial marker with the PA-GFP signal.

We believe the root of the reviewer's confusions lies in our explanation of our experiments, and we have amended the text to more clearly reflect our methodology. To summarize – we conducted two separate experiments to evaluate the effect of the actin wave on mitochondrial content mixing. First, in interphase cells we labeled mitochondrial subpopulations with photoactivatable GFP and recorded the percent change in PA-GFP-positive area, which is agnostic to how motile the mitochondria are. A decrease in this metric reflects reduced fusion of distinct mitochondria, and we found such a decrease upon wave ablation (Fig 5B). Our second experiment used manual tracking to record the percentage of mitochondria that undergo heterofusion as a result of the wave compared to baseline (sampled vs. unsampled), and we found this percentage greater for the former group (Fig 5C). The results of both experiments therefore support our hypothesis that the wave promotes mitochondrial content mixing.

Reviewer 3

The actin wave in Fig. 1A is difficult to spot: perhaps a differential or ratiometric analysis between subsequent images would better highlight where the dynamics are?

We have now annotated the images to better indicate the position of the actin wave.

The actin "wave" in Fig. 2C (control) does not circulate as the authors suggest is the typical pattern. Please comment on this – is this non-circular trajectory common?

(Now refers to Fig 2D) We occasionally observe non-circular trajectories for actin waves. These seem to represent the wave migrating above or below the nucleus, and therefore momentarily exiting the focal plane.

How was TMRE quantified? If absolute intensities were used, they can be misleading due to differential uptake: this is mitigated by normalizing against a different probe.

To evaluate the degree of TMRE loading in cells treated with either control oligo or FMNL1 siRNA we first expressed mito-sBFP2, a matrix marker largely independent of polarization

status. Then after adding TMRE and washing out excess, we randomly imaged cells. For quantification we used the mito-sBFP2 channel to segment mitochondria and then determined, for each cell, the average mitochondrial TMRE signal. While it is possible that cells differentially take up TMRE, we are unaware of literature that indicates this, and further any skewing effect would be mitigated by our large sample size. It is also worth noting that the methodology and analysis we have utilized here has been widely employed in the field (for example, Shi et al., Nature Communications 2022).

How the quantitative analysis of actin wave (area determination, speed determination) is done is unclear, a section on this with a schematic figure would help, and code should be made available.

In our revised manuscript we have included a schematic illustrating how our actin wave area and speed measurements were taken (Fig S1A), and we discuss these points more clearly in the methods section. All code utilized is able to be fully described in the methods section.

Analysis of photoconversion expansion: The authors use Ilastik and difference in area of thresholded objects. The spreading is low during the timeframe used, and using intensity values instead of thresholding would be more sensitive. For example, analyzing the variance of the intensity distribution at different timepoints. I also found the image display (green on grey) confusing, as grey looks like a dim green.

Regarding analysis of our photoconversion experiment, we used signal intensity in order to threshold. Therefore, perhaps the reviewer's primary concern is best addressed by clarifying our explanation of our methodology, which we have done in the revised manuscript in the methods section. Further, it may be useful to note that our approach here mirrored that used by others in the field (for example, Molina & Shirihai, Methods Enzymol 2009).

In response to the second point, we have changed the color scheme for the images in order to make key features more perceptible.

Fig 2B: TMRE uptake is variable, as seen by the two populations (low and high). A justified exclusion of the statistical outliers from the analysis would make sense, or normalising (as in previous point).

The break in the Y axis, with unequal scaling between the two segments, could have led to the misunderstanding that the TMRE data is characterized by two populations. Actually, long tails are seen here. In the revised manuscript we have more explicitly noted the unconventional Y axis.

SEM is used instead of SD, this should be justified.

Throughout the manuscript whenever SEM is used it is when individual data points are displayed. Such data transparency allows for an even fuller assessment of the degree to which data points differ from means than SDs afford. We chose to overlay SEMs in these cases to provide additional information – namely an estimate of how far sample means are from population means.

Statistical tests used (one-way ANOVA and student's t test) assume normal distribution of data, which in this case it doesn't seem to be. Normality tests and Mann-Whitney and Kruskal-Wallis one-way ANOVA tests should be performed.

We note that we have taken care to choose the correct statistical test for each comparison in the manuscript. While for some comparisons it was appropriate to use parametric tests, for others normally distributed data could not be assumed and so non-parametric tests were

employed (for example, Fig 1C, Fig S2E, and Fig 5C). We have reviewed the manuscript to ensure the specific tests used are noted within each figure legend.

Some figures need more cells to support the statistics (e.g. Fig 5C)

We have now utilized power analysis for experiments with lower sample sizes, identified ones which require more measurements, and made appropriate adjustments. Specifically, we found our analysis of actin wave speed after adavosertib treatment to be under-powered, and we corrected this by observing more cells (Fig 2G), with the expanded data ultimately indicating the same conclusion. We further identified our experiment assessing cellular ATP levels as requiring a greater sample size. After collecting more data we could not detect a statistically significant effect, noting too much variability between measurements, and have thus removed this experiment from the manuscript. Finally, we replaced our figure in the main text assessing PA-GFP spread after CK-666 administration (previously Fig 5C) with a parallel experiment assessing this same metric but after FMNL1 knock-down, as this intervention is more specific in ablating the actin wave (Fig 5B). Here power analysis indicates a higher sample size may be ideal, though achieving this is logistically infeasible and already n is increased 2.6X.

Videos would benefit from slower playback for ease of seeing the wave.

We have adjusted the videos accordingly.

Fig. 6A is missing the quantification of size (+/-actin) for untreated cells. Figure S3B: This would be the corresponding plot to Fig. 6, but it lacks mito size quantification for (+/-actin)

We agree that adding this quantification paints a fuller picture of the relationship between the actin wave, microtubules, and mitochondrial dynamics. Thus, we now provide the missing quantification (Fig 6A).

Figure 6B should color code the time, to allow the reader to better observe the dynamics.

We appreciate the reviewer's insightful suggestion and believe incorporating it would allow our images to convey further, important information. Thus, we have now altered the figure accordingly.

The study is performed on a single cell line, HeLa. The authors should comment in the discussion about how universal they expect their findings to be. They have previously published limited data on other cell types (NHEK, HaCaT, HEK 293T, COS-7, and iPS). However, the wave in many of those was not as striking and, in some cases, barely discernible to this reviewer. It does not diminish the interest of the work, but other researchers in the field would benefit from understanding the expected limitations versus universality of the findings.

In the revised manuscript we now present our findings from key experiments performed using another cell line, COS-7. Specifically, we now demonstrate in this alternative cell line that FMNL1 knock-down ablates the actin wave (Fig S1L-N), that FMNL1 knock-down reduces mitochondrial membrane potential (Fig S3A), that the actin wave can promote mitochondrial content mixing (Fig S4A), and finally that upon microtubule depolymerization wave-associated actin comet tail mitochondrial motility can be observed (Fig S5C). We believe these data lend credence to the idea that the findings of our manuscript are applicable to different cell types, and we now make this point throughout the paper while acknowledging in the discussion section that the full scope of cell types to which our findings apply to remains to be investigated.

Lines 444-447: This sentence doesn't make sense to me.

(Now refers to lines 512-514) In the revised manuscript we had edited the sentence for clarity.

Line 415: The link between mitochondrial size and metabolic state is correlative and not causal, to my knowledge.

(Refers to lines 482-484 in revised manuscript) We have now edited the text to reflect the larger point we hoped to make here while taking care not to over-state any science that remains unsettled.

Figure 2E: Wording of legend is confusing.

(Now refers to figure 2N) In the revised manuscript we have amended the legend for clarity.

Figure 4A: Typo in Lifeact label

We thank the reviewer for their careful reading of our manuscript and have now corrected this typo.

Summary of Revisions: Again, we thank all three referees for their thoughtful critiques. Addressing each of these points has made our work both stronger and clearer. We hope that this work will now be found acceptable for publication in *Nature Communications*.

REVIEWER COMMENTS

Reviewer #1 (Remarks to the Author):

In the revised manuscript, Coscia et al. improve the rigor of the work and address many of the points I raised in my original review. However, I have the following remaining comments and concerns that should be addressed in order to recommend publication:

1. The authors now provide additional supportive data that FMNL1 is a CDK1 substrate, including showing slightly increased phosphorylation of FMNL1 in the presence of an inhibitor of a CDK1 inhibitor. However, the evidence provided remains indirect. A more convincing line of evidence would be to demonstrate reduced phosphorylation of FMNL1 in CDK1 knockdown cells or in the presence of CDK1 inhibitor. The authors should also show reduced phosphorylation of the S1013A variant.
2. The effect of the S1013A and S1013E mutant variants on actin waves is not clear in the images shown (Fig. 2K, 2O). In Figure 2K, the images appear to show more actin colocalized with mitochondria in GFP-S1013A than in GFP-WT. Further, if S1013A is dominant-negative, as proposed by the authors, why does the actin staining fail to phenocopy staining of FMNL1 and CDK1 KD (Fig. 1B and 2F)?
3. It also appears in Fig. 2O that cells adjacent to the ones depicted have widely varying amounts of sub-cortical actin staining. It is not clear from the figure legend how many cells are quantified in this experiment, whether data from multiple independent experiments are included, and how cells were picked for analysis. Are the 20% of cells with no actin wave that are found in wild type cells (Fig. 1D) omitted? For transparency, the box and whisker plots should be changed to show individual data points (as in Fig. 2N, 2R).
4. While it is understandable the authors chose to not set up a technically challenging complementation assay, the authors should remove claims that actin waves promote complementation from the abstract and introduction. The TMRE and seahorse assays do demonstrate effects on mitochondrial homeostasis, which is accurately reflected in the title. The proposed role in complementation would instead be more appropriate to raise in the discussion.

Reviewer #2 (Remarks to the Author):

The reviewers have address all of my concerns.

Reviewer #3 (Remarks to the Author):

The revised version of the manuscript by Coscia et al. addressed my main critiques, and I find it to be suitable for publication. The findings are truly exciting, and although I have some minor technical quibbles, I understand the authors' points that they are generally following the practices in the field.

Response to Review, *Coscia et al.*

We very much appreciate the thoughtful consideration of our revised manuscript, and are pleased that Referees 2 and 3 now find our work suitable for publication and “truly exciting.” We also thank Reviewer 1 for providing further suggestions for improving the manuscript, which we now address as described in the point-by-point response below; revisions from the first round are highlighted in yellow in the text while revisions from this second round of review are highlighted in green. It is our hope that with these additional minor revisions our work will now be found acceptable for publication in *Nature Communications*.

Reviewer 1

The authors now provide additional supportive data that FMNL1 is a CDK1 substrate, including showing slightly increased phosphorylation of FMNL1 in the presence of an inhibitor of a CDK1 inhibitor. However, the evidence provided remains indirect. A more convincing line of evidence would be to demonstrate reduced phosphorylation of FMNL1 in CDK1 knockdown cells or in the presence of CDK1 inhibitor. The authors should also show reduced phosphorylation of the S1031A variant.

We agree that examining FMNL1 phosphorylation levels in response to CDK1 inhibition would further test our hypothesis that these two are a kinase-substrate pair. In fact, we had already performed the first experiment suggested here.

As shown in Panel A, and in support of our hypothesis, we found that in 4 out of 5 biological replicates of this experiment FMNL1 phosphorylation was decreased in response to the CDK1 inhibitor Ro-3306, as assessed using an antibody specific for the phosphorylated CDK1 consensus sequence. Unfortunately, however, due to experimental variability with the immunoprecipitations and limits to the sensitivity of the phospho site-specific antibody, these data did not reach statistical significance. Thus we have not included these these data in our manuscript.

We also performed the second suggested experiment, comparing the phosphorylation levels of WT and S1031A FMNL1 (panel B). Our data show that FMNL1 S1031A exhibits less reactivity to the antibody specific for phosphorylated CDK1 substrate consensus sequence, consistent with our hypothesis. Still, we prefer not to include these data

CDK1 inhibition and FMNL1 S1031A mutation likely decreases CDK1 consensus site phosphorylation on FMNL1. A) GFP-FMNL1 expressing samples were treated with either DMSO or the CDK1 inhibitor Ro-3306 (20 μ M, 2.5hr), and GFP was immunoprecipitated with resulting eluate analyzed by Western blot. Representative blot shown with accompanying quantitation, where different dots represent different biological replicates and different colors represent slightly different versions of protocol. Note, for purple dots, two are magenta and one is dark purple – indicating it corresponds to representative blot. B) GFP-WT FMNL1 or GFP-FMNL1 S1031A samples were subjected to GFP pull-down and Western blot analysis. Representative blot shown with accompanying quantitation.

A Replicate with **medium** decrease in phos signal (dark purple in graph)

B

in order to avoid making a potentially incorrect assumption – it is unclear if the antibody reacts with equivalent affinities to the WT phosphorylated site and the phosphomimetic mutant we have generated.

It is important to note, however, that there is significant literature indicating that CDK1 phosphorylates FMNL1 at S1031, and we now do a better job of summarizing these findings in the revised text. First, Petrone and colleagues (Mol Cell Proteomics 2016) compared the phospho-proteome of control and CDK1 inhibitor-treated cells to find that FMNL1 is differentially phosphorylated at S1031. Also, PhosphoSitePlus (Hornbeck et al., Nucleic Acids Res. 2012), a curated database of phosphorylation events observed with mass spectrometry, contains several entries of experiments where analytes were isolated by pull-down with an antibody specific for the phosphorylated CDK substrate consensus sequence and noted phosphorylation of FMNL1 at S1031. Thus, we believe that the entirety of our data, in conjunction with the relevant published literature which is now more fully described, strongly support the hypothesis that CDK1 phosphorylates FMNL1 at S1031. However, in response to the point raised we have softened our conclusions on this point in the revised manuscript.

The effect of the S1013A and S1013E mutant variants on actin waves is not clear in the images shown (Fig. 2K, 2O). In Figure 2K, the images appear to show more actin colocalized with mitochondria in GFP-S1013A than in GFP-WT. Further, if S1013A is dominant-negative, as proposed by the authors, why does the actin staining fail to phenocopy staining of FMNL1 and CDK1 KD (Fig. 1B and 2F)? We sincerely thank the reviewer for catching a mistake in our figure – in the course of the first round of revisions, the representative images for the WT and FMNL1 S1031A conditions were inadvertently switched. We have now corrected Fig. 2K, and have double checked all figures throughout the manuscript for accuracy. These data make it clear that FMNL1 S1031A overexpression does indeed phenocopy FMNL1 and CDK1 knock-down.

It also appears in Fig. 2O that cells adjacent to the ones depicted have widely varying amounts of sub-cortical actin staining. It is not clear from the figure legend how many cells are quantified in this experiment, whether data from multiple independent experiments are included, and how cells were picked for analysis. Are the 20% of cells with no actin wave that are found in wild type cells (Fig. 1D) omitted? For transparency, the box and whisker plots should be changed to show individual data points (as in Fig. 2N, 2R).

Again, we appreciate the reviewer's attentive read of our manuscript. We note that in the previous set of images only the center, outlined cell was entirely within the field of view, and therefore it is not possible to assess the actin wave in the other, peripheral cells. To more clearly convey our finding without this distraction, we have now re-cropped the images to better focus attention on the representative cells of interest. Importantly, we provide full descriptions of the sample size, number of biological replicates, and method of choosing cells for imaging/analysis within the text accompanying Figure 2O, reinforcing the previous discussion of these points within the relevant figure legend and within the Methods section. Finally, we now show data points in box and whisker plots as suggested. Please note that for the box and whisker plots in Figure 2H, the DMSO controls were performed in parallel with the CDK1 inhibitors as part of a larger screen; the control data for this screen were previously published in Moore et al. (2021) as noted in the figure legend, but the CDK1 inhibitor data were not previously published and are thus reported here.

While it is understandable the authors chose to not set up a technically challenging complementation assay, the authors should remove claims that actin waves promote complementation from the abstract and introduction. The TMRE and seahorse assays do demonstrate affects on mitochondrial homeostasis, which is accurately reflected in the title. The proposed role in complementation would instead be more appropriate to raise in the discussion.

We agree that while our data indicate that the interphase actin wave mediates mitochondrial content mixing and homeostasis, we do not independently test complementation. Thus, we have modified our language as requested throughout the revised manuscript.

REVIEWERS' COMMENTS

Reviewer #1 (Remarks to the Author):

The authors have addressed my remaining concerns and I recommend publication of the work.